# Marine Debris in the Beilun Estuary Mangrove Forest: Monitoring, Assessment and Implications

**DOI:** 10.3390/ijerph182010826

**Published:** 2021-10-15

**Authors:** Dongmei Li, Li Zhao, Zhiming Guo, Xi Yang, Wei Deng, Haoxiang Zhong, Peng Zhou

**Affiliations:** 1South China Sea Environment Monitoring Center, State Oceanic Administration (SOA), Guangzhou 510300, China; lidmay@foxmail.com (D.L.); ronda1522@163.com (L.Z.); guozhiming0917@163.com (Z.G.); yang1209xi@163.com (X.Y.); dw42968166@163.com (W.D.); 2South China Sea Testing and Appraisal Center, State Oceanic Administration (SOA), Guangzhou 510300, China; 3Nansha Islands Coral Reef Ecosystem National Observation and Research Station, Guangzhou 510300, China; 4Key Laboratory of Marine Environmental Survey Technology and Application, Ministry of Natural Resources (MNR), Guangzhou 510300, China; 5Guangzhou Institute of Energy Conversion, Chinese Academy of Sciences, Guangzhou 510640, China

**Keywords:** debris pollution, mangrove forest, Beilun Estuary, monitoring and assessment

## Abstract

A modified approach for marine debris investigation in mangrove forests is developed, including some practical programs, viz., sampling location, time, area, materials, size and sources data processing. The marine debris method was practiced in the Beilun Estuary mangrove forest region in Fangchenggang in 2019, viz., the debris items were classified, counted, weighed and recorded, and the marine debris pollution was assessed to understand the impact of human activities. The results show that the mass density is 21.123 (2.355~51.760) g/m^2^, and more than 90% came from the land-based and human activities. More than 60% of the total debris weights are plastics, followed by fabrics (17.91%) and Styrofoam (10.07%); the big-size and oversize debris account for 76.41% and 13.33%, respectively. The quantity density is 0.163 (0.013~0.420) item/m^2^, and ~95% came from land-based human activities. More than 75% of the total debris items were plastics, followed by Styrofoam (14.36%), fabrics (4.10%) and glass (3.59%); the big-size, medium-size and oversize debris are 76.41%, 13.33% and 10.26%, respectively. The results suggest that mangrove forests are barriers for the medium-/big-size marine debris, acting as traps for marine debris. Our study provides recommendations and practical guidance for establishing programs to monitor and assess the distribution and abundance of marine debris. The results show that mangrove areas in the Beilun Estuary are filled with some plastic debris (plastics plus Styrofoam) and that the density and type at Zhushan and Rongshutou near the China-Vietnam border are more than those at Shijiao and Jiaodong. The results of this study are also expected to not only provide baseline data for the future assessment of Beilun Estuary mangroves but also to help China and Vietnam strengthen marine land-based pollution control and promote coastal wetland and mangrove conservation, marine species conservation and sustainable use.

## 1. Introduction

A mangrove is one of the distinctive woody plants in the coastal wetland ecosystem with both terrestrial and marine properties, which can provide a variety of important goods and services to humanity [1,2,3,4,5,6], e.g., absorbing the waves/tides, protecting the shore, maintaining the biodiversity, accelerating the water purification and pollutants degradation, reducing the eutrophication and red tide, developing ecological tourism and popular science education. Mangroves are highly susceptible to marine debris (litter) exposure due to their coastal habitats [7]. Marine debris (litter) was defined by UNEP in 1995 as: ‘any persistent, manufactured or processed solid material discarded, disposed of or abandoned in the marine and coastal environment’ [8]. Marine debris (including plastics) is introduced into the marine environment by its improper disposal, accidental loss and natural disasters. Martin et al. (2019) [9] demonstrated that mangrove forests act as sinks for anthropogenic debris before they are dispersed into the marine environment. In the last few decades, marine debris has been recognized as an indicator of pollution forms, causing risks to marine organisms [8,10]. Marine debris pollution has been a focal point of public concern and a visible expression of the human impact on the marine environment [11,12]. Similarly, the mangrove ecosystem is threatened by marine debris pollution at present, including visual pollution and toxic substances (heavy metals, organic pollutants, pathogenic organisms, etc.) carried by the debris itself as well as via ingestion by marine organisms. It is urgent to monitor marine debris and to assess the debris coverage and their impact on mangroves.

For the marine debris survey, some methods have been successfully established for marine debris monitoring and assessment in the marine environment (surface seawater, seabed and beaches) [8,13,14,15,16,17]. The approaches/methods comprise the visual method, net method (trawl) and diving method [12,15]. Moreover, some innovative approaches here include satellite remote sensing [18,19,20], the aerial photography method [20,21,22] using coupled balloon-assisted photos with in situ mass measurements [23], ortho-photograph from planes [19] and several initiatives using drones [24]. These innovative approaches are particularly useful for detecting larger litter items in dense vegetation (e.g., reed beds), for non- destructive observations in sensitive habitats (e.g., salt marshes) and for remote or inaccessible coastlines [8]. Ongoing monitoring activities can be used to assess the effectiveness of the management strategies and provide insight when strategies need to be modified for changing conditions. There was a lack of harmonization of the sampling methods and attention to natural environmental variability, so this made the collation and comparison of the data problematic [8,13]. The implementation of effective control measures is currently hampered by a lack of the consistent monitoring and identification of the sources of the debris [12].

No appropriate and special approach is used for monitoring and assessing marine debris in mangrove forests. Mangroves are usually located in the wetland and marsh areas of estuaries and are subjected to tidal influences, so the debris investigation in mangrove forests (especially in different inter-tidal zones) is much more difficult than in seawater, seabed and beaches. The satellite and photography methods are not applicable because of the canopy of the mangrove branches, so the visual and weight methods are still simple and practical. Although the marine debris survey in mangrove forests is similar to those in sea water, beach and seabed [8,12,13,14,15], there are some differences. Up to now, there are a few studies on the marine debris monitoring and assessment in mangroves [9,25], but there is no appropriate and special method. The marine debris pollution (including plastics and microplastics) in mangrove forest regions is actually affected by both natural factors (i.e., hydrodynamics, mangrove height and density, etc.) and human activities (mariculture, tourism and coastal dumping) [26,27,28,29]. Thus, it is essential to develop an appropriate method of marine debris monitoring in mangroves to provide effective recommendations and practical guidance, and to conduct surveys to assess the impact of human activities on coastal zones.

The Beilun Estuary mangrove forest reserve region is located at the southern offshore areas of Gangkou District and the city of Dongxing of the city of Fangchenggang, China, bordered by the Beibu Gulf in the southeast and Vietnam in the southwest, and spanning Pearl Bay, Jiangping Estuary, and Beilun Estuary, with a coastline of 105 km and a tidal flat area of 53 km^2^ [29,30,31,32,33]. Here, we provide a modified approach for marine debris investigation in mangrove forests, including some practical programs, viz., sampling location, time, area, materials, size and sources data processing. The marine debris method was applied and practiced in the Beilun Estuary mangrove forest region in 2019, viz., the debris items were classified, counted, weighted, recorded and assessed for the distribution and abundance of the marine debris pollution. It is hoped that our study is useful for providing recommendations and practical guidance for monitoring and assessing marine debris in mangrove forests, and for understanding the basic information of the mangrove ecological threat factors and to assess the influence of human activities. This study is also expected to not only provide baseline data for the future assessment of the marine debris pollution in Beilun Estuary mangroves but also help China and Vietnam strengthen marine land-based pollution control and promote coastal wetland and mangrove conservation, marine species conservation and sustainable use.

## 2. Methodology

### 2.1. Method Establishment

One approach to make best use of limited resources is to take advantage of other studies and programs where litter monitoring can be integrated [8]. By referring to the current methods or guidelines for marine debris (in sea water, seabed and beach) monitoring and assessment [8,13,14,15], a special approach for marine debris in mangrove forests is developed using the visual and weight methods. For a specific procedure, some practical programs are established in brief. Firstly, sampling location (transect and station) and time are drawn up according to the survey purpose. Secondly, sampling area should be determined according to the actual surrounding conditions (debris density, mangrove density, sludge hazard degree, tidal time, etc.). Thirdly, the debris items are in situ collected and categorized by material types. Some information of each debris item, such as weight, sizes and sources, should be recorded. It is necessary to take photos or video on site and record the other information. Finally, all debris should be collected and removed as far as possible in order to protect the mangrove living environment. More details of marine debris investigation are as follows:

(1)Sampling location, time and area

Sampling location (transect and station) and time setting can refer to the general specifications or guidelines of mangrove survey or marine inter-tidal zone. Moreover, sampling location and time setting must consider the surrounding environment conditions and the flood and ebb tides. Sampling should not always take place at a constant location/transect. Each transect includes three stations, viz., the high-tide station, the medium-tide station and the low-tide station. Random sampling location can also be used. Sampling time must comply with the safety regulations and the tide time because of the complex environment (mud everywhere, twining branches). If the tide rises fast, sampling time duration must be controlled to ensure the personnel security before the rising tide. Additionally, personal protection (to prevent mosquitoes and snake bites) must be prepared; monitoring work must stop when it rains and lightning strikes.

The selection of sampling area (m^2^) in each station generally follows the following principles: the larger sampling area is, the more representative and reliable the results are. Moreover, the densities of the debris and mangrove must be considered, as well as the surrounding environment conditions and duration between the flood and ebb tides. Debris coverage was estimated in 10 m × 10 m block, which is divided into 1 m^2^ quadrats, and 20 quadrats from the 100 m^2^ block were sampled in each site [34]. For ease of operation, if a comprehensive mangrove ecological investigation is carried out, the location, time and area are same as or close to those of debris in the mangrove forests. Here, the area in each station is often 100 m^2^. If the density is too small, the area can be appropriately increased. If the density is very high or the environment is unfavorable for the investigation, the area can be reduced but should not be less than 5 m × 5 m.

(2)Materials, size and sources

Table 1 provides marine debris classification based on the size, materials and sources. Here, one difference from other methods is that the disposable paper cups (beverage cartons) coated with single-sided PE plastic film or double-sided PE plastic film are regarded as plastic debris because the remaining plastic film after paper fiber being damaged easily will disperse lots of serious secondary microplastics into the environment. Papers refer to paper products without plastic coating that are big, degradable and endanger the organisms in the mangrove forests. So, the papers that are very easy to degrade in the slush are not considered. Moreover, wood products refer to mainly those old furniture and ships that are large, hinder the landscape and may endanger the mangrove forest growth. Here, the wood sticks and furniture fragments that are small and do not endanger the mangrove growth are not considered.

The details of the materials/compositions (e.g., plastic, glass, metal, etc.), the overall form/shape (e.g., bottle, film, rope, net, bag, etc.) and the size should be recorded as possible during the investigation. These details are very useful because the properties of the debris can affect the behavior of the debris in the environment, including the further degradation, transport and the extent and nature of any impacts. Moreover, they can be very useful, especially for providing information about the relative importance and potential sources or other specific policy concern, including the effectiveness of targeted reduction measures [8,35]. Usually, macro-litter items will offer more clues as to their origins since they can be more easily associated with their original use [8]. The ocean-borne waste disposed at sea and terrestrial waste originating from coastal users and urban centers are two main sources [36,37,38,39,40].

(3)The density estimation and assessment

The density is calculated for the number (items) or mass (weights, g) of debris in each category per the area (A) as:(1)$$D=\frac{N\mathrm{umber}\;(\mathrm{items})\;\mathrm{or}\;\mathrm{Mass}\;(g)}{A\mathrm{rea}}$$
where D is the quantity density in items per 1.0 square meters (items/m^2^) or mass density in terms of grams per 1.0 square meters (g/m^2^), respectively. Additionally, for the entire mangrove region, the total quantity (items) and mass (kg) in each type of debris, e.g., plastics, are estimated based on the total area of mangrove region.

Our approach is suitable for all visual debris items encountered in mangrove ecosystem, deposited on the seabed and/or associated with encrusted/entangled on the branches/roots. This approach can also provide effective assurance for harmonization of sampling/monitoring methods, identification of sources, attention to natural environmental variability, collation and comparison of data. It can not only be used alone for the special marine debris survey but also be used in conjunction with the comprehensive mangrove ecosystem investigation. However, there are still some disadvantages, e.g., large personnel workload and high manpower consumption (compared with remote sensing method).

### 2.2. Application and Practice in Beilun Estuary Mangrove Region

In Fangchenggang, Guangxi Zhuang Autonomous Region, there are 580-km coastline and ~7500-hectares coastal wetland, including 3300 hectares of mangrove reserve: Beilun Estuary national mangrove forest reserve region [30,31]. As one of the largest mangrove forests in the world, Beilun Estuary mangrove forest has ~1300 hectares, referred to as green lungs by the Fangchenggang citizens. However, with the aggravation of human disturbance in the coastal zone and global climate change, the green lungs face both natural and anthropogenic threats [41], viz., introduction of pollutants (e.g., oils, metals and sewage) [42,43,44], over-fishing and harvesting, habitat loss and fragmentation (reclamation, urban infrastructure and construction) [41,45], invasion of alien species [31,33,46], pests and diseases [41], microplastic pollution [29,47,48] and man-made debris.

Marine debris in four transects of mangrove forest (Shijiao, Jiaodong, Zhushan and Rongshutou) was investigated in September 2019 (Figure 1). Only the high tide and low tide stations were investigated in Rongshutou because of its short transect. Three stations (the high-tide, the medium-tide and the low-tide stations) were investigated in Shijiao, Jiaodong and Zhushan transects (Table 2). Figure 2 provides some photos in mangrove forest.

## 3. Results and Discussion

Table 3, Figure 3 and Figure 4 provide the mass and quantity densities and percentages of the debris in the Shijiao, Jiaodong, Zhushan and Rongshutou transects, respectively. The debris in the Shijiao transect consisted of plastics, Styrofoam and glasses. Only two types (both plastics and Styrofoam) were found in the Jiaodong transect. In the Zhushan transect, four types of debris were found, viz., plastics, Styrofoam, fabrics and glasses, and one electronic debris (a mobile phone) was also found. In the Rongshutou transect, six types of debris (including plastics, Styrofoam, fabrics, glasses, metals and rubbers) were found. In the Beilun Estuary mangrove forest region, the mass densities of the debris ranged from 2.355 to 51.760 g/m^2^, with the mean value of 21.123 g/m^2^. The quantity densities of the debris ranged from 0.013 to 0.420 item/m^2^, with the mean value of 0.163 item/m^2^, which was relatively lower than that in the estuary of Mandonga and Lahundap [25].

Table 4 provides the source of the marine debris in the Beilun Estuary mangrove forest region. For the marine debris, the mass percentage of 9.19% and the quantity percentage of 5.64% came from marine activities, viz., navigation/fishing activity. In the land-based debris, the mass percentage of 0.12% and the quantity percentage of 1.54% came from the medical or sanitary activity. More than 90% (both mass and quantity percentages) came from the coastal/recreational activity. More than 50% of the land-based debris were plastics, followed by Styrofoam (more than 10%).

### 3.1. Space Distribution of Marine Debris

The mass density of the debris in Zhushan is the highest (51.760 g/m^2^), followed by in Rongshutou (29.200 g/m^2^), Shijiao (10.617 g/m^2^) and Jiaodong (2.355 g/m^2^). The mass density in Zhushan is ~25 times as high as that in Jiaodong. The quantity density of the debris in Zhushan is the highest (0.420 item/m^2^), followed by in Rongshutou (0.197 item/m^2^), Shijiao (0.024 item/m^2^) and Jiaodong (0.013 item/m^2^), with a difference of about 50 times between the maximum and minimum values. In four transects, the order of the debris mass/quantity densities is as follows: Zhushan > Rongshutou > Shijiao > Jiaodong. The types of debris in the Rongshutou and Zhushan transects are more than those in the Shijiao and Jiaodong transects. Some used clothes were also found in both Rongshutou and Zhushan. Therefore, a great deal of marine debris comes from human activities.

The two sections of Rongshutou and Zhushan are very close to China’s Dongxing and Vietnam’s Mangjie urban areas. They have a large population and have especially developed tourism. Due to poor management or other reasons, a large amount of garbage enters the mangrove forest along the Beilun River. At the Zhushan and Rongshutou transects near the China-Vietnam border, the mass and quantity densities and types of debris were much more than those at other transects. It shows the importance of two countries jointly controlling the garbage discharge and reducing the marine debris pollution. The marine debris management and control should be the responsibility of both China and Vietnam. In Figure 2, the debris in situ collected indicated manufacturers’ marks and addresses on labels from either China or Vietnam. The labels indicate the place of manufacture but not necessarily where an item entered an environmental compartment. For example, litter from ships may originate far from where it is discarded and be transported before reaching the sampling location (UNEP, 2016) [35]. The labels are enough to prove that marine debris management is actually an international problem.

In order to protect the mangrove forests and reduce the debris, more efforts had been made in Fangchenggang, China. The production, sale and use of ultra-thin bags (<0.025 mm) was banned in China on 1 June 2008. The Household Garbage Classification Plan has been implemented since 2017 [49]. Moreover, China implemented a new policy banning the importation of most plastic waste in 2018 [50]. Meanwhile, a specific plan of household garbage classification work has been formulated and implemented in the city of Fangchenggang [51]. In another recent document [52], the ban of the production and sale of ultra-thin plastic bags and agricultural films (<0.01 mm) will remain strictly enforced; at the same time, our government will ban the importation of plastic waste, as well as the production and sale of single-use foam plastic tableware and plastic swabs by the end of 2020; for daily chemicals containing plastic microbeads, production will be banned this year, while sales must stop by 2022. Although some policies and measures have been formulated, this is still not enough. Both China and Vietnam need to make greater efforts to strengthen waste management and reduce waste emissions.

### 3.2. Density and Total Debris Weight in Beilun Estuary Mangrove Forest Region

In the Beilun Estuary mangrove forest region, the mean mass and quantity densities are 21.123 g/m^2^ (2.355~51.760) and 0.163 item/m^2^. The total estimated weights of the debris range from ~30.6 to ~672.9 tons, with a mean value of ~274.6 tons (Table 5). Such a large amount of debris should not be ignored because the debris of plastics, foam, glass, tires and electronic waste in situ may contaminate the mangrove environment inevitably. The mass percentage of plastics in all the debris is the highest (63.10%), followed by fabrics (17.91%) and Styrofoam (10.07%). The quantity percentage of plastics is the highest (75.90%), followed by Styrofoam (14.36%), fabrics (4.10%) and glasses (3.59%). The marine debris may potentially threaten the health and safety of marine life (birds, dolphins, sea turtles, etc.) by being ingested, entangling and suffocating. Additionally, the debris may affect the coastal landscape and have adverse effects on human health and the marine economy.

### 3.3. Size of the Debris in Beilun Estuary Mangrove Forest Region

Figure 4 provides the mass and quantity percentages of the debris by size in the mangrove forest (Table 6 and Figure 5). The mass percentages of the big-size, oversize and medium-size debris are 72.18%, 26.67% and 1.16%, respectively, while the quantity percentages are 76.41%, 10.26% and 13.33%, respectively. The small-size debris is negligible in the Beilun Estuary mangrove forest region. Additionally, more than 98% of the debris are harmful, or could be harmful (including plastics, foam, textile materials and some rubber), which is another source of microplastic debris. The medium-/big-size debris (plastics, fabrics and rubber) may ultimately degrade into millions of microplastic particles because of the mechanical stress, sun exposure and weathering conditions. Microplastics are of environmental concern because their size (millimeters or smaller) renders them accessible to a wide range of organisms at least as small as zooplankton, with the potential for physical and toxicological harm [53]. Microplastic pollution in the mangrove sediments was investigated in Qinzhou and Fangchenggang, Guangxi [29,47,48]. If conditions permit, it will be necessary to investigate microplastics in seawater and sediments in mangroves as much as possible in the future.

### 3.4. Debris Distribution Influenced by Flood-Ebb Fluctuations

The debris distribution in mangrove forests may be related to the flood-ebb tidal current. The debris can enter a mangrove forest during the rising or falling tide and may be lodged by branches or roots. The maximum mass density of the marine debris is at the low-tide or medium-tide station of the Jiaodong, Shijiao and Zhushan transects, while it is at the high-medium-tide station of the Rongshutou transect; the minimum mass density is at the high-tide station in the Shijiao, Jiaodong and Zhushan transects, while it is at the low-tide station in the Rongshutou transect. Additionally, the maximum quantity density of the marine debris is at the low-tide station in the Zhushan transect; the maximum quantity density is at the medium-tide station in the Shijiao transect, while it is at the high- or medium-tide station in the Rongshutou transect. In the Jiaodong, Zhushan and Rongshutou transects, the minimum quantity density is at the low- or medium-tide station.

No small-size debris were found, but large debris (the medium-/big-size/oversize) were found in the mangrove forest region. These debris are mainly light-weighted, floating plastic debris (plastic bags and Styrofoam). Our results support that mangrove forests are traps [9] for the medium-/big-size, land-/ocean-based floating debris. Additionally, our results suggest that the pneumatophores of a mangrove forest act as a filter, which can not only prevent the land-based debris from entering the marine environment through the river but also prevent the sea-based debris from being dispersed into the terrestrial environment again by tidal currents and waves [9,26]. We agree with the viewpoint that mangroves are akin to sinks for marine debris. Figuratively speaking, mangroves act as a barrier, blocking the land-/ocean-based debris from dispersing.

### 3.5. Plastic Debris Pollution and Assessment

Plastic, as an important basic material, is widely used in production and life. Although modern life would be unrecognizable without plastics, the dark side of the convenience they bring must not be ignored [54], including the entanglement of marine fauna, ingestion by seabirds and organisms ranging in size from plankton to marine mammals, dispersal of microbial and colonizing species to potentially non-native waters and concentration and transport of organic contaminants to marine organisms at multiple trophic levels [55]. The magnitude and fate of plastic debris pollution are still open questions [56,57,58] because of their persistence at sea (their durability, low recycling rates) and adverse consequences to marine life and potentially human health [16,37,59]. Mangroves have been recognized for their importance in confining plastic waste due to their pneumatophores. The spatial complexity characteristic of mangrove ecosystems provides many opportunities for debris to become entangled, which enhances their role as traps and/or barriers for plastic pollution dispersing [7,9,60].

If Styrofoam is also considered as plastic debris in this study, the percentage of the total plastic debris is more than 70% (the mass percentage of 73.17% and the quantity percentage of 90.25%). The mass densities of plastics and Styrofoam are 13.329 and 2.128 g/m^2^, respectively; the quantity densities are 0.123 and 0.023 item/m^2^, respectively. The total input weights of plastics and Styrofoam debris are 30.3~506.2 tons and 0.2~95.2 tons, with the mean values of ~173.3 tons and ~27.7 tons, respectively. Such a large amount of plastic debris not only pollutes the environment but may also transform into microplastics in the future, which will inevitably harm marine life in the mangrove environment. Thus, plastic (plastics and Styrofoam) products should not be used excessively, and, at the same time, plastic waste should not be discharged but reused as much as possible. Only curbing plastic use and reducing waste can save our planet. More mandatory policies should be formulated and implemented to prompt market players to reduce packaging materials and increase the rates of recycling and reuse, as well as establishing a green certification system for the industry [52].

## 4. Conclusions

The magnitude and fate of this pollution are still open questions. An appropriate modified approach of the marine debris in mangroves was established and practiced in the Beilun Estuary mangrove region. Our results show that mangroves act mostly as a barrier for the big-size debris because it can prevent both the land-based debris from dispersing into the sea and the ocean-based debris from invading the terrestrial environment. It is hoped that our methodology may be generalized for other regions in the world, and that it is helpful to promote the standardized monitoring of marine debris in the mangrove ecosystem to assess the impact of human activities on mangroves. This study is also expected to help promote biodiversity-friendly practices and approaches for the conservation and sustainable use of threatened ecosystems and species in the mangrove ecosystem.

The vast mangrove ecosystem in the Beilun Estuary provides the food and habitat for a variety of benthos and birds. We must clearly recognize that the mangrove areas in the Beilun Estuary are filled with some plastic debris (plastics plus Styrofoam: more than 70% Ms-p.c and 90% Qt-p.c.) due to human activities. In the Beilun Estuary mangrove region, marine debris pollution is actually an international problem as the Beilun Estuary region is a land connection for economic cooperation and logistics between China and ASEAN. Dongxi in Fangchenggang (China) is only separated from Mong Cai (Vietnam) by the Beilun River. With the acceleration of industrialization and urbanization and the rapid development of the economy and society, the mangroves are facing serious threats due to urban expansion, coastal development, high-intensity human interference and other factors. Our results are helpful to promote the supervision and control of marine debris, especially plastics, in the mangrove region of the Beilun Estuary of China and Vietnam. The findings of this study provide the baseline data for marine debris, including plastic pollution, and assist in prioritizing future plastic debris monitoring and mitigation strategies.

## Figures and Tables

**Figure 1 ijerph-18-10826-f001:**
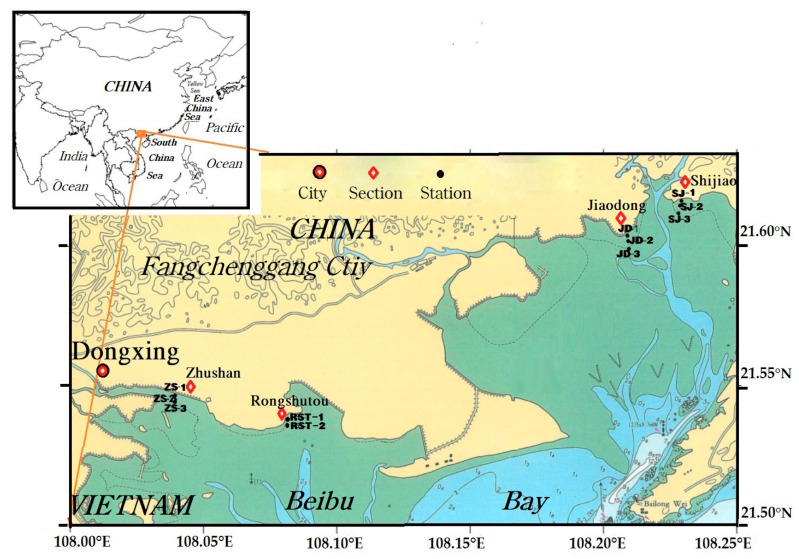
Transects and stations of the debris monitoring in Beilun Estuary mangrove forest region.

**Figure 2 ijerph-18-10826-f002:**
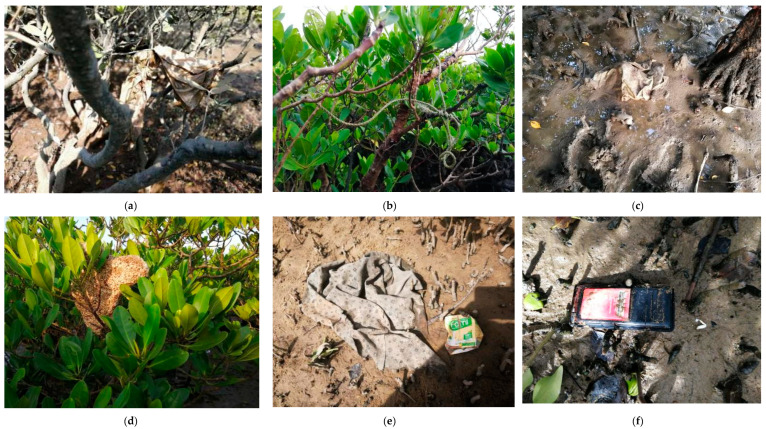
The in situ monitoring and collection of the debris in mangrove forest. (**a**) Plastic bag hung by the branch in Jiaodong; (**b**) Rope hung by branches in Jiaodong; (**c**) Plastic bag in the mud at Jiaodong; (**d**) Beverage boxes and used clothes at Rongshutou; (**e**) Styrofoam hung by the branch in Zhushan; (**f**) Electronic debris in Zhushan; (**g**) In situ collection of the debris.

**Figure 3 ijerph-18-10826-f003:**
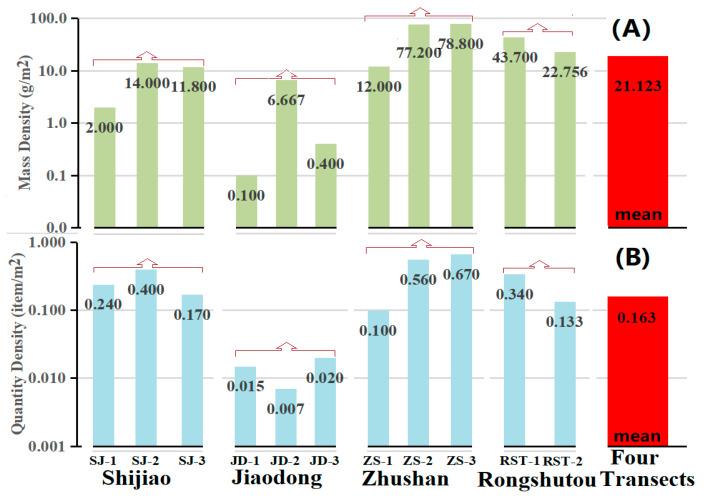
The mass (**A**) and quantity (**B**) densities of the debris.

**Figure 4 ijerph-18-10826-f004:**
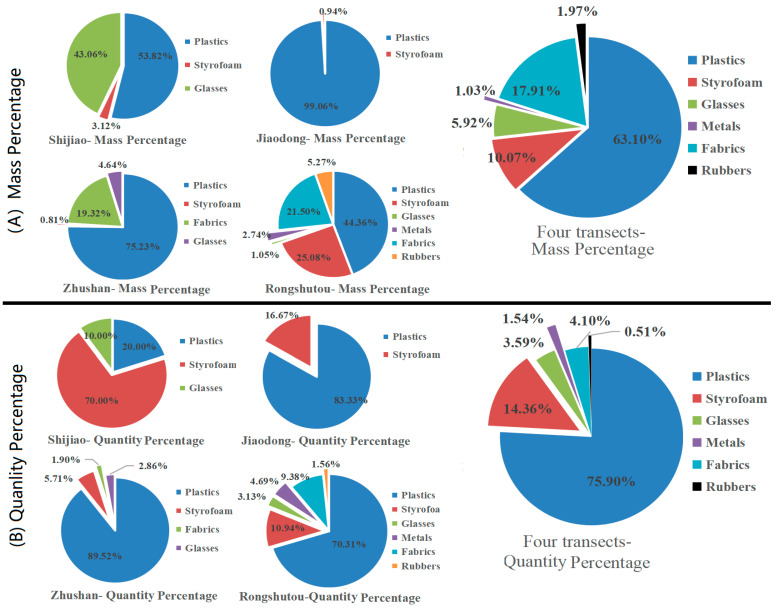
The mass (**A**) and quantity (**B**) percentages of the debris.

**Figure 5 ijerph-18-10826-f005:**
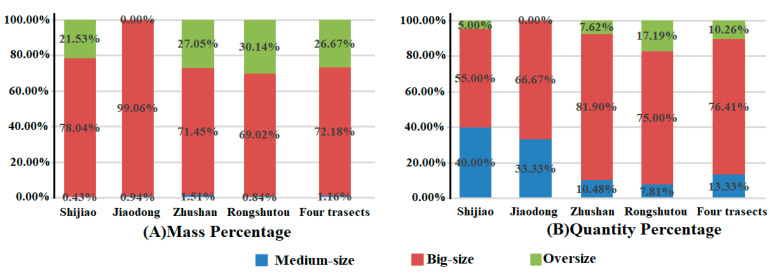
The percentages of medium-size, big-size and oversize debris.

**Table 1 ijerph-18-10826-t001:** List and classification of the debris based on the size, materials and sources.

Classification		Specific Examples
Materials/ Composition (plastics, Styrofoam, wood, paper, metal, rubber, fabric/fiber, glass, other material) *	Plastics	Bags, bottles, cigarette filters, lighters, buckets, lid, spoons, knives and forks, straws, hats, diapers, syringes, fishing lines, nets, floats, safety helmets, feeding bottles, ropes, toys, rings, old plastic boat, etc.;
	Styrofoam	Buoys, cups, foam boxes, fast food boxes (plates), etc.;
	Glasses	Bottles, fluorescent tubes, bulbs, glass fragments, etc.;
	Metals	metal barrels, beverage cans, metal plates, metal fragments, iron wires, etc.;
	Rubbers	balloons, rubber gloves, tires, condoms, etc.;
	Fabrics	Clothes, rags and textile materials, etc.;
	Wood products ^#^	Furniture, crates, chopsticks, wooden cases, old wooden boat, etc.;
	Papers ^$^	Paper bags, cardboards, cups, newspapers, etc.;
	Others	Other artificial articles and unrecognizable materials
Size	Small-size (<2.5 cm), Medium-size (≥2.5 cm and ≤10 cm), Large-size (>10 cm and ≤1 m) or Oversize (>1 m) debris	
Sources	Sea sources	coastal/recreational activity, navigation/fishing activity, medical or sanitary activity and other disposal source (e.g., wood, ceramics, plastic pieces, glass slices)
	Land sources	

Note: * The relative persistency: food waste < paper < wood < iron < plastic. ^#^ Wood products are mainly those furniture and ships that are large, hinder the landscape and may endanger the mangrove forest growth. ^$^ Papers refer to paper products without plastic coating that are big, degradable and endanger the organisms in the mangrove forests. The disposable paper cups (or beverage cartons) are considered as plastics because the paper surface is coated with single- or double-sided PE or plastic film.

**Table 2 ijerph-18-10826-t002:** The debris monitoring stations in Beilun Estuary mangrove forest region.

Transect	Date	Station		Area (m^2^)	Debris	Station of Rising and Falling Tide
Shijiao	28 August 2019	SJ-1	108.2282° E, 21.6169° N	5 × 5	Foam particles	High-tide station
		SJ-2	108.2284° E, 21.6152° N	10 × 5	Beer bottle, plastic woven bag	Medium-tide station
		SJ-3	108.2283° E, 21.6142° N	10 × 10	Beer bottle, broken buoy, plastic reticule, mineral water bottle, foam particles	Low-tide station
Jiaodong	28 August 2019	JD-1	108.2042° E, 21.6028° N	15 × 10	Foam, plastic cup	High-tide station
		JD-2	108.2053° E, 21.6016° N	15 × 10	Plastic woven bag	Medium-tide station
		JD-3	108.2053° E, 21.6000° N	10 × 10	Plastic bag, rope	Low-tide station
Zhushan	29 August 2019	ZS-1	108.0399° E, 21.5469° N	10 × 10	Small/big plastic bag	High-tide station
		ZS-2	108.0396° E, 21.5469° N	10 × 5	Plastic packing bag, woven bag, big plastic bag, mineral water bottle, plastic disposal cup, foam	Medium-tide station
		ZS-3	108.0382° E, 21.5468° N	10 × 10	Canvas, pants, beer bottle, woven bag, small/big plastic bag, plastic bag, fruits packing net, plastic cup/drink bottle/food box, foam fast food box, packing rope, pipe	Low-tide station
Rongshu tou	30 August 2019	RST-1	108.0806° E, 21.5388° N	10 × 10	Foam, small plastic bag, glass bottle, ring-pull can, motorbike pedal (metal), plastic drug board, used clothes, shoes, rubber tyre, rope, cellphone	High-medium-tide station
		RST-2	108.0878° E, 21.5375° N	15 × 15	fishing nets, plastic film, foam, metal food bottle, woven bag, used clothes, cotton gloves	Low-tide station

**Table 3 ijerph-18-10826-t003:** Mass weights and quantities of debris in Shijiao, Jiaodong, Zhushan and Rongshutou transects in Beilun Estuary mangrove forest.

Types	D-Mass (g/m^2^)	Ms-p.c. (%)	D-Qt (item/m^2^)	Qt-p.c. (%)	Stations	Types	Area (m^2^)	Weights (g)	D-Ms (g/m^2^)	Ms-p.c. (%)	Item	D-Qt (item/m^2^)	Qt-p.c. (%)
Shijiao transect (Station of SJ-1, SJ-2, and SJ-3 )Area = 175 m^2^					SJ-1	Styrofoam	25	50	2.000	100	6	0.240	100
					SJ-2	Plastics	50	300	8.000	57.14	1	0.020	50.00
Styrofoam	0.331	3.12	0.080	70.00		Glasses		400	6.000	42.86	1	0.020	50.00
Plastics	5.714	53.82	0.023	20.00		D-Mean		-	14.000	-	-	0.040	-
Glasses	4.571	43.06	0.011	10.00	SJ-3	Styrofoam	100	8	0.080	0.72	8	0.080	66.67
D-Mean	10.617	-	0.024	-		Plastics		600	6.000	54.15	3	0.030	25.00
						Glasses		500	5.000	45.13	1	0.010	8.33
						D-Mean		-	11.800	-	-	0.120	-
Jiaodong transect (Station JD-1, JD-2, and JD-3), Area = 450 m^2^					JD-1	Styrofoam	200	10	0.050	50.00	1	0.005	33.33
						Plastics		10	0.050	50.00	2	0.010	66.67
Styrofoam	0.022	0.94	0.002	16.67		D-Mean		-	0.100	-	-	0.015	-
Plastics	2.333	99.06	0.011	83.33	JD-2	Plastics	150	1000	6.667	100.00	1	0.007	100.00
D-Mean	2.355	-	0.013	-	JD-3	Plastics	100	40	0.400	100.00	2	0.020	100.00
Zhushan transect (Station ZS-1, ZS-2, and ZS-3), Area = 250 m^2^					ZS-1	Plastics	100	1200	12.000	100.00	10	0.100	100.00
					ZS-2	Styrofoam	50	80	1.600	50.00	4	0.080	50.00
Styrofoam	0.420	0.81	0.024	5.71		Plastics		3780	75.600	50.00	24	0.480	50.00
Plastics	38.940	75.23	0.376	89.52		D-Mean		-	77.200	-	-	0.560	-
Fabrics	10.000	19.32	0.008	1.90	ZS-3	Styrofoam	100	25	0.250	0.32	2	0.020	2.99
Glasses	2.400	4.64	0.012	2.86		Plastics		4755	47.550	60.34	60	0.600	89.55
D-Mean	51.760	-	0.420	-		Fabrics		2500	25.000	31.73	2	0.020	2.99
						Glasses		600	6.000	7.61	3	0.030	4.48
						D-Mean		-	78.800	-	-	0.670	-
Rongshutou transect (Station RST-1 and RST-2), Area = 425 m^2^					RST-1	Styrofoam	100	2300	23.000	52.63	2	0.020	5.88
						Plastics		1010	10.100	23.11	24	0.240	70.59
Styrofoam	7.323	25.08	0.022	10.94		Glasses		100	1.000	2.29	2	0.020	5.88
Plastics	12.954	44.36	0.138	70.31		Metals		160	1.600	3.66	2	0.020	5.88
Glasses	0.308	1.05	0.006	3.13		Fabrics		300	3.000	6.86	3	0.030	8.82
Metals	0.800	2.74	0.009	4.69		Rubbers		500	5.000	11.44	1	0.010	2.94
Fabrics	6.277	21.50	0.018	9.38		D-Mean		-	43.700	-	-	0.340	-
Rubbers	1.538	5.27	0.003	1.56	RST-2	Plastics	225	3200	14.222	62.50	21	0.093	70.00
D-Mean	29.200	-	0.197	-		Styrofoam		80	0.356	1.56	5	0.022	16.67
						Metals		100	0.444	1.95	1	0.004	3.33
						Fabrics		1740	7.733	33.98	3	0.013	10.00
						D-Mean		-	22.756	-	-	0.133	-
Four transects (Shijiao, Jiaodong, Zhushan, and Rongshutou), Area = 1200 m^2^					Total	Styrofoam	1200	2553	2.128	10.07	28	0.123	14.36
						Plastics		15,995	13.329	63.10	148	0.023	75.90
						Glasses		1500	1.250	5.92	7	0.006	3.59
						Metals		260	0.217	1.03	3	0.003	1.54
						Fabrics		4540	3.783	17.91	8	0.007	4.10
						Rubbers		500	0.417	1.97	1	0.001	0.51
						D-Mean		25,348	21.123	-	195	0.163	-

Note: D-Ms and D-Qt represent mass density (g/m^2^) and quantity density (item/m^2^), respectively. Ms-p.c (%) and Qt-p.c (%) represent the percentage of mass and quantity, respectively. D-Mean represents mean density (g/m^2^) of (item/m^2^).

**Table 4 ijerph-18-10826-t004:** Source of marine debris in Beilun Estuary mangrove forest region.

Sources		Types	Ms-p.c. (%)	Qt-p.c. (%)	Debris Items
Land-based	Coastal/Recreationalactivity	Styrofoam	10.07%	14.36%	Particles, foam fast food box, etc.
		Plastics	53.79%	68.72%	Plastic film, plastic bag (woven bag, plastic reticule, packing bag), plastic bottle/box/cup (mineral water bottle, drink bottle/food box, disposal cup), fruits packing net, packing rope, pipe, cellphone, etc.
		Glasses	5.92%	3.59%	Beer bottle, glass bottle/cup
		Metals	1.03%	1.54%	ring-pull can, metal food bottle, motorbike pedal
		Rubbers	17.91%	4.10%	Rubber tire
		Fabrics	1.97%	0.51%	Used clothes, shoes, pants, cotton gloves
		Total	90.69%	92.82%	-
Land-based	Medical or sanitary activity	Plastics	0.12%	1.54%	Plastic drug board
Sea-based	Navigation/fishing activity	Plastics	9.19%	5.64%	Fishing nets, broken buoy, rope

**Table 5 ijerph-18-10826-t005:** The debris weights estimated in Beilun Estuary mangrove forest region.

Area	1200 m^2^ of Survey Area		~1.3 × 10^7^ m^2^ of Mangrove Area	
Classification	D-Ms (g/m^2^)		Debris Weights (ton)	
	Mean	Range	Mean	Range
Plastics	13.329	2.333~38.940	173.3	30.3~506.2
Styrofoam	2.128	0.022~7.323	27.7	0.3~95.2
Fabrics	3.783	0~10.00	49.2	0~130.0
Rubbers	0.417	0~1.538	5.4	0~20.0
Metals	0.217	0~0.800	2.8	0~10.4
Glasses	1.250	0~2.400	16.2	0~31.2
All debrs	21.123	2.355~51.760	274.6	30.65~672.9

**Table 6 ijerph-18-10826-t006:** Mass weights and quantity of marine debris items by sizes.

Transect		Mass Weights (g)			Quantity (Item)		
		Medium-Size	Big-Size	Oversize	Medium-Size	Big-Size	Oversize
Shijiao	Weight/Quantity	8	1450	400	8	11	1
	p.c (%)	0.43	78.04	21.53	40.00	55.00	5.00
Jiaodong	Weight/Quantity	10	1050	0	2	4	0
	p.c (%)	0.94	99.06	0.00	33.33	66.67	0.00
Zhushan	Weight/Quantity	195	9245	3500	11	86	8
	p.c (%)	1.51	71.45	27.05	10.48	81.90	7.62
Rongshutou	Weight/Quantity	80	6550	2860	5	48	11
	p.c (%)	0.84	69.02	30.14	7.81	75.00	17.19
Four transects	Weight/Quantity	293	18,295	6760	26	149	20
	p.c (%)	1.16	72.18	26.67	13.33	76.41	10.26

## Data Availability

The data presented in this study are available in insert article here.

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
