# Peer review of "Marine Debris in the Beilun Estuary Mangrove Forest: Monitoring, Assessment and Implications"

_ijerph, 2021, doi:10.3390/ijerph182010826_

Round 1

Reviewer 1 Report

This paper introduced the status of marine debris in the Beilun estuary mangrove forest. This study is very helpful for understanding the environment status and making the future plan for protecting the environment in the study area. The author called the proposed approach " the visual and weight method for marine debris in mangrove forest", but did not provide details about what exactly the approach is. Please provide more details about the approach and how the experiments were conducted. Please fix the typos (e.g., Addationally -> Additionally) and polish the language as well.

Author Response

Reviewer #1:

Author's Reply to the Review Report (Reviewer 1)

Open Review

English language and style

( ) Extensive editing of English language and style required
(x) Moderate English changes required
( ) English language and style are fine/minor spell check required
( ) I don't feel qualified to judge about the English language and style

Yes

Can be improved

Must be improved

Not applicable

Does the introduction provide sufficient background and include all relevant references?

(x)

( )

( )

( )

Is the research design appropriate?

(x)

( )

( )

( )

Are the methods adequately described?

( )

(x)

( )

( )

Are the results clearly presented?

(x)

( )

( )

( )

Are the conclusions supported by the results?

(x)

( )

( )

( )

Comments and Suggestions for Authors

This paper introduced the status of marine debris in the Beilun estuary mangrove forest. This study is very helpful for understanding the environment status and making the future plan for protecting the environment in the study area.

  1. The author called the proposed approach " the visual and weight method for marine debris in mangrove forest", but did not provide details about what exactly the approach is. Please provide more details about the approach and how the experiments were conducted.

R:

OK.

Section of “2. Materials and methods” is reorganized.

  • The procedure ofthe visual and weight method is briefly described according to the steps of the investigation.

2.1. Method Establishment

One approach to make best use of limited resources is to take advantage of other studies and programmes where litter monitoring can be integrated [8]. By referring to the current methods or guidelines for marine debris (in sea water, seabed, and breach) monitoring and assessment[8,13-15], an special approach for marine debris in mangrove forest is developed using the visual and weight methods. For a specific procedure, some practical programmes are established in belief: Firstly, sampling location (transect and station) and time are drawn up according to the survey purpose. Secondly, sampling area should be determined according to the actual surrounding conditions (debris density, mangrove density, sludge hazard degree, tidal time, etc.). Thirdly, the debris items are in-situ collected and categorized by material types. Some information of each debris item, such as weight, sizes, and sources, should be recorded. It is necessary to take photos or video on site and record the other information. Finally, all debris should be collected and removed as far as possible in order to protect the mangrove living environment. More attention details of marine debris investigation are as follows:

(1) Sampling location, time and area

Sampling location (transect and station) and time setting can refer to the general specifications or guidelines of mangrove survey or marine inter-tidal zone. Also, sampling location and time setting must consider the surrounding environment conditions and the flood and ebb tides. Sampling should not always take place at a constant location/transect.  Each transect include three stations, viz., the high-tide station, the medium-tide station, and the low-tide station. Random sampling location can also be used. Sampling time must comply with the safety regulations and the tide time because of the complex environment (mud everywhere, twining branches). If the tide rises fast, sampling time duration must be controlled to ensure the personnel security before the rising tide. Additionally, personal protection (to prevent mosquitoes and snake bites) must be prepared; monitoring work must stop when it rains and lightning strikes.

The selection of sampling area (m2) in each station generally follows the following principles: The larger sampling area is, the more representative and reliable the results are. Also, the densities of the debris and mangrove must been considered, as well as  the surrounding environment conditions and Duration between the flood and ebb tides. Debris coverage was estimated in 10 m×10 m block which is divided into 1 m2 quadrats and 20 quadrats from the 100 m2 block were sampled in each site [34]. For ease of operation, if a comprehensive mangrove ecological investigation are carried out, the location, time and area are same as or close to those of debris in the mangrove forsts. Here, the area in each station are often 100 m2. If the density is too small, the area can be appropriately increased. If the density is very high or the environment is unfavorable for the investigation, the area can be reduced but should not be less than 5 m×5 m.

(2) Materials, size and sources

Table 1 provides marine debris classification based on the size, materials and sources. Here, one difference from other methods is that the disposable paper cups (beverage cartons) coated with single-sided PE plastic film or double-sided PE or plastic film are regarded as plastic debris. Because the remaining plastic film after paper fiber being damaged easily will form lots of serious secondary microplastics into the environment. Moreover, wood products refer to mainly those old furniture and ships that are large, hinder the landscape, and may endanger the mangrove forest growth. Here, the wood sticks and furniture fragments that are small and do not endanger the mangroves growth are not considered.

The details of the materials/compositions (e.g., plastic, glass, metal, etc.), the overall form/shape (e.g., bottle, film, rope, net, bag, etc.), and the size should be recorded as possible during the investigation. It is very useful because their properties can affect the debris behaviors in the environment, including the further degradation, transport and the extent and nature of any impacts. Moreover, they can be very useful, especially for proving information about the relative importance and potential sources, or other specific policy concern, including the effectiveness of targeted reduction measures [8,35]. . Usually macro-litter items will offer more clues as to their origins, since they can be more easily associated with their original use [8]. The ocean borne waste disposed at sea and terrestrial waste originating from coastal users and urban centers are two main sources [36-40].

(3) The density estimation and assessment

The density is calculated for the number (items) or mass (weights, g)  of debris in each category per the area (A) as:

      (1)

whereis the quantity density in items per 1.0 square meters (items/m2) or mass density in terms of grams per 1.0 square meters (g/m2), respectively. Additionally,for the entire mangrove region, the total quantity (items) and mass (kg) in each type of debris, e.g., plastics, is estimated based on the total area of mangrove region.

  • Someadvantages of our method proposed here is also described.  

Our approach is suitable for all visual debris items encountered in mangrove ecosystem, deposited on thOur approach is suitable for all visual debris items encountered in mangrove ecosystem, deposited on the seabed, and/or associated with encrusted/entangled on the branchs/roots. This approach can also provide effective assurance for harmonization of sampling/monitoring methods, identification of sources, attention to natural environmental variability, collation and comparison of data. It can not only be used alone for the special marine debris survey, but also be used in conjunction with the comprehensive mangrove ecosystem investigation. However, there are still some disadvantages, eg., large personnel workload and high manpower consumption (compared with remote sensing method).

  1. Please fix the typos (e.g., Addationally -> Additionally) and polish the language as well.

R:

   OK.

  • In Line 133 OFPage 3/14, Line 260 of Page 8/13, Line 267, 285 and 294 of Page 9/13, Line 345 of Page 10/13, the word “Addationally” has been changed into “Additionally”.

Others,

  • In Line 69 of Page 1/14, Line 344 of Page 10/14, the word of “harmonisation”has been changed into “harmonization” .
  • In Line 81 of Page 2/14, In Table 1, Line 143 of Page 4/14, and In Line 259 of Page 8/14, the word of “etc”was changed into “”
  • In Line 127 of Page 3/14, Line 359 of Page 11/14, the word of “enviroment”has been changed into “environment” .
  • In Line 140 of Page 3/14, Line 359 of Page 11/14, the word of “investigatinon”has been changed into “investigation” .
  • In Line 192 of Page 5/14, Line 359 of Page 11/14, the word of “mairne”has been changed into “marine” .
  • In Line 214-215 of Page 6/14, in Figure 2 the words of “(d) Plastic bag and box at Rongshutou”  has been changed into “(d) Beverage boxes and used clothes at Rongshutou” .
  • In Line 216-217 of Page 6-7/14, in Table 2,
  • the words of “small/Big plastic bag”,“clothes”, “ cloths” has been changed into “small/big plastic bag”, “used clothes” .
  • In Line 358-359 of Page 11/14, the word of “recommendation”has been changed into “recommendations” .
  • In Line 362.367 of Page 11/14, the word of “Fangcheng”has been changed into “Fangchenggang city” .

We modify other parts as follows

  1. In Line 147 of Page 4/14, “Note”in Table 1 has been added as follows:

Classification

Specific examples

Materials/Composition

(plastics, styrofoam, wood, paper, metal, rubber, fabric/fiber,glass, other material)*

Plastics

Bags, bottles, cigarette filters, lighters, buckets, lid, spoons, knives and forks, straws, hats, diapers, syringes, fishing lines, nets, floats, safety helmets, feeding bottles, ropes, toys, rings, old plastic boat, etc.;

Styrofoam

Buoys, cups, foam boxes, fast food boxes (plates), etc.;

Glasses

Bottles, fluorescent tubes, bulbs, glass fragments, etc.; 

Metals

metal barrels, beverage cans, metal plates, metal fragments,iron wires, etc.;

Rubbers

balloons, rubber gloves, tires, condoms, etc.;

Fabrics

Clothes, rags, and textile materials, etc.;

Wood products#

Furniture, crates,chopsticks,wooden cases, old wooden boat, , etc.;

Papers$

Paper bags, cardboards, cups, newspapers, etc.;

Others

Other artificial articles and unrecognizable materials

Size

Small-size (<2.5 cm), Medium-size (≥2.5 cm and ≤10 cm), Large-size (>10 cm and ≤1 m) or Oversize (>1 m) debris

Sources

Marine sources

coastal/recreational activity, navigation/fishing activity, medical or sanitary activity, and other disposal source (e.g. wood, ceramics, plastic pieces, glass slices)

Land sources

Note:*The relative persist-ency: food waste < paper < wood < iron < plastic. # Wood products are mainly those furniture and ships that are large, hinder the landscape, and may endanger the mangrove forest growth. $Papers refer to paper products without plastic coating. The disposable paper cups (or beverage cartons) are considered as plastics, because the paper surface is coated with single- or double-sided PE or plastic film.

  1. SI Table 1  has been changed into “Table 3.

1 ) “SI Table 1 ” has been changed into “Table 3. Mass weights and quantities of debris in Shijiao, Jiaodong, Zhushan and Rongshutou transects in Beilun estuary mangrove forest”.

2) The content pattern of the table has been reproduced and adjusted.

3) Some Errors in the table have been modified, e.g., marine debris item, quantity density (item/m2) and Percentage(%) in Zhushan transect (Station ZS-1, ZS-2, and ZS-3 ), the mumblers of “0.200” for Fabrics, “0.612” “3.92”, “61.44”, “32.68”, “1.96” have been changed into “0.008” , “0.420” “5.71”, “89.52”, “1.90”, “2.86” , the number of “50” has been deleted (should be “2”).

 Some Errors in Total-Four transects (Shijiao , Jiaodong , Zhushan, and Rongshutou) have been also modified:  the mumblers of “0.047”  for Fabrics , “0.203” , “11.52”, “60.91”, “2.88”, “1.23” , “23.05” , “0.41”  have been changed into “0.007 ” , “0.163”,  “14.36”, “75.90”, “3.59”, “1.54” , “4.10” , “0.51”  , the number of “56” has been deleted (should be “8”).

In “Figure 3. The mass and quantity densities and the percentages of the debris”, “Abstract”, and  “Results and Discussion”,  the corresponding errors have been modified.

  1. Table 4. Source of marine debris in Beilun estuary mangrove forest region has been added.  In Results and Discussion,  the corresponding errors have been added as follows:

Table 4 provides source of marine debris in Beilun estuary mangrove forest region. For marine debris, the mass percentage of 9.19% and the quantity percentage of 5.64% came from marine activities,viz., navigation/fishing activity. In the land-based debris, the mass percentage of 0.12% and the quantity percentage of 1.54% came from the medical or sanitary activity. More than 90% (both mass and quantity percentages) came from the coastal/recreational activity. More than 50% of the land-based debris were plastics, followed by styrofoam (more than 10%).

  1. SI Table 2  has been changed into “Table 6.

Reviewer 2 Report

The manuscript provides data on the types/amounts of debris found in a specific mangrove forest in China.  My main challenge with this manuscript is not the data provided or the methods used-those seem logical.  What I don't understand is the introduction and conclusions made on the data.  From the intro it appears the paper will discuss how a rational method can and should be developed, but that is not described in the experimental section- just what methods were done but not how they came to be.  The conclusions also talk about the development of an appropriate method- but it is unclear what the development was as opposed to just doing a method.  Additional focus in the introduction is needed with more appropriate conclusions.  If the paper was changed to an evaluation of debris in mangrove as opposed to development of a sampling method for debris in mangrove, I think that makes more sense.

Author Response

Reviewer #2:

Author's Reply to the Review Report (Reviewer 2)

Open Review

English language and style

(x) Extensive editing of English language and style required
( ) Moderate English changes required
( ) English language and style are fine/minor spell check required
( ) I don't feel qualified to judge about the English language and style

Yes

Can be improved

Must be improved

Not applicable

Does the introduction provide sufficient background and include all relevant references?

( )

( )

(x)

( )

Is the research design appropriate?

( )

( )

(x)

( )

Are the methods adequately described?

( )

( )

(x)

( )

Are the results clearly presented?

(x)

( )

( )

( )

Are the conclusions supported by the results?

( )

( )

(x)

( )

Comments and Suggestions for Authors

  1. The manuscript provides data on the types/amounts of debris found in a specific mangrove forest in China.My main challenge with this manuscript is not the data provided or the methods used-those seem logical.What I don't understand is the introduction and conclusions made on the data. 

R:

  1.  The sections of “Introductionand“Conclusions” have been rewritten.
  • The content pattern of Introductionhas been reproduced and adjusted. Line 85-line 109, Page 2-3/14, the first and second paragraphs of Methodology have been transferred to the introduction. Moreover, in order to indicate clear the study objective, the end of the introduction has been rewritten. Some parts have been deleted.

  The Introduction has been modified as follows:

Mangrove is one of the distinctive woody plants in the coastal wetland ecosystem with both terrestrial and marine properties, which can provide a variety of important goods and services to humanity[1-6], e.g., absorbing the waves/tides, protecting the shore, maintaining the biodiversity, accelerating the water purification and pollutants degradation, reducing the eutrophication and red tide, developing the ecological tourism and popular science education. Mangroves are highly susceptible to marine debris (litter) exposure due to their coastal habitats[7]. Marine debris (litter) was proposed by UNEP in 1995 as: ‘any persistent, manufactured or processed solid material discarded, disposed of or abandoned in the marine and coastal environment’[8]. Marine debris (including plastics) is introduced into marine environment by its improper-disposal, accidental loss, and natural disasters. Martin et al. (2019)[9] demonstrated that mangrove forest act as sinks for anthropogenic debris before they are dispersed into the marine environment. In the last few decades, marine debris has been recognized as an indicator of pollution forms causing risks to marine organisms [8,10]. Marine debris pollution has been a focal point for public concern and a visible expression of human impact on the marine environment [11,12]. Similarly, mangrove ecosystem is threatened by marine debris pollution at present, including the visual pollution, the toxic substances (heavy metals, organic pollutants, pathogenic organisms, etc.) carried by the debris itself, as well as the ingestion of marine organisms. It is urgent to monitor marine debris and to assess the debris coverage and their impacts on mangroves.

For marine debris survey, some methods have been successfully established for marine debris monitoring and assessment in marine environment (surface seawater, seabed, and beaches)[8, 13-17]. The approaches/methods comprise the visual method, net method (trawl), diving method [12, 15]. Also, some innovative approaches here include satellite remote sensing [18-20], aerial photography method [20-22] using coupled balloon-assisted photos with in situ mass measurements [23], ortho-photograph from planes[19], and several initiatives using drones [24]. These innovative approaches are particularly useful for detecting larger litter items in dense vegetation (e.g. reed beds), for non- destructive observations in sensitive habitats (e.g. salt marshes) and for remote or inaccessible coastlines [8]. Ongoing monitoring activities can be used to assess the effectiveness of management strategies, and provide an insight when strategies need to be modified for changing conditions. There was a lack of harmonization of sampling methods and attention to natural environmental variability, so that this made the collation and comparison of data problematic [8,13]. The implementation of effective control measures is currently hampered by a lack of consistent monitoring and identification of sources of debris [12].

No appropriate and special approach is used for monitoring and assessing marine debris in mangroves forest. Mangroves are usually located in wetland and marsh areas of estuaries and are subjected to tidal influences, so that the debris investigation in mangrove forests (especially in different inter-tidal zones) is much more difficult than those in seawater, seabed, and beaches. The satellite and photography method is not applicable because of the canopy of the mangrove branches, so that the visual and weight methods are still simple and practical. Although marine debris survey in mangrove forest is similar to that in sea water, beach and seabed [8,12-15], there is some differences. Up to now, there are a few studies on marine debris monitoring and assessing in mangroves [9,25], but there is no appropriate and special method. Marine debris pollution (including plastics and microplastics) in mangrove forest regions are actually affected by both natural factors (i.e., hydrodynamics, mangrove height and density, etc.) and human activities (mariculture, tourism and coastal dumping)[26-29]. Thus, it is essential to develop an appropriate method of marine debris in mangroves monitoring, to provide effective recommendations and practical guidances, and to conduct surveys to assess the impact of human activities on coastal zones.

Beilun estuary mangrove forest reserve region is located at the southern offshore areas of Gangkou district and Dongxing city of Fangchenggang city, China, bordered Beibu gulf in the southeast and Vietnam in the southwest, and spanned Pearl bay, Jiangping estuary, and Beilun estuary, with a coastline of 105 km and a tidal flat area of 53 km2 [29, 30-33]. Here we provide an approach for marine debris investigation in mangrove forest including some practical programmes, viz., sampling location, time, area, materials, size and sources, data processing. Marine debris method was applied and practiced in Beilun estuary mangrove forest region in 2019, viz., the debris items were classified, counted, weighted, recorded, and assessed the distribution and abundance of marine debris pollution. It is hoped that our study is useful for providing recommendations and practical guidance for monitoring and assessing marine debris in mangrove forest, and for understanding the basic information of mangrove ecological threat factors and to assess the influence of human activities. This study is also expected to not only provide baseline data for the future assessment of marine debris pollution in Beilun estuary mangroves, but also help China and Vietnam strengthening marine land-based pollution control, and promote coastal wetland and mangrove conservation, marine species conservation and sustainable use.

  • The content pattern of Conclusionshas been reproduced and adjusted  in order to indicate clear the study objective. Moreover, the last paragraph has been changed into “ 1. Space distribution of marine debris” in 3. Results and Discussion——In order to protect mangrove forest and reduce the debris, more efforts has been made in Fangcheng city, China. The production, sales and use of ultra-thin bags (<0.025mm) was banned in China on June 1, 2008. Household Garbage Classification Plan has been implemented since 2017 (NDRC and MHUD, P.R. Cna, 2017). Also, China implemented a new policy banning the importation of most plastic waste in 2018 (Brooks et al., 2018). Meanwhile, a specific plan of household garbage classification work has been formulated and implemented in Fangcheng city (PGGZAR, P.R. China, 2017). In another new document recently(NRC and MEE, P.R. China, 2020), the ban of the production and sale of ultra-thin plastic bags and agricultural films (<0.01 mm) will remain strictly enforced; at the same time, our government will ban imports of plastic waste, as well as the production and sales of single-use foam plastic tableware and plastic swabs by the end of 2020; for daily chemicals containing plastic microbeads, production will be banned this year, while sales must stop by 2022.

The Section of Conclusions” has been modified as follows:

“The magnitude and the fate of this pollution are still open questions. A appropriate approach of marine debris in mangrove was established and practiced in the Beilun estuary mangrove region. Our results show that mangroves act mostly as a barrier of the big-size debris, because it can prevent both the land-based debris from dispersing into the sea and the ocean-based debris from invading the terrestrial environment. It is hoped that our methodology may be generalized for other regions in the world, and that it is helpful to promote the standardized monitoring of marine debris in mangrove ecosystem to assess the impact of human activities on mangroves. This study is also expected to help biodiversity friendly practices and approaches promoted for conservation and sustainable use of threatened ecosystems and species in mangrove ecosystem.

The vast mangrove ecosystem in the Beilun estuary provides food and habitat for a variety of benthos and birds. We must clearly recognize that mangrove areas in Beilun estuary are filled with some plastic debris ( plastics plus styrofoam: more than 70% Ms- p.c and 90% Qt-p.c.) due to human activities. In Beilun estuary mangrove region, marine debris pollution is actually an international problem. Because the Beilun estuary region is a land connection for economic cooperation and logistics between China and ASEAN. Dongxi in Fangchenggang (China) is only separated from Mong Cai (Vietnam) by the Beilun River. With the acceleration of industrialization and urbanization, and the rapid development of economy and society, the mangroves are facing serious threats due to urban expansion, coastal development, high-intensity human interference, and other factors. Our results is helpful to promote the supervision and control of marine debris, especially plastics, in the mangrove region of the Beilun estuary of China and Vietnam. The findings of this study provide baseline data for marine debris including plastic pollution, and assist in prioritizing future plastic debris monitoring and mitigation strategies.”

2.From the intro it appears the paper will discuss how a rational method can and should be developed, but that is not described in the experimental section- just what methods were done but not how they came to be. 

R:

OK.

Section of “2. Materials and methods” has been rewritten.

1)The title of the section of “2. Materials and methods” has changed into 2. Methodology .

2) The experimental section 2.1. Method Establishment” also been rewritten in detail.

For a specific procedure for marine debris investigation in mangrove forest, some practical programmes are described in general language: “Firstly, sampling location (transect and station) and time are drawn up according to the survey purpose. Secondly, sampling area should be determined according to the actual surrounding conditions (debris density, mangrove density, sludge hazard degree, tidal time, etc.). Thirdly, the debris items are in-situ collected and categorized by material types. Some information of each debris item, such as weight, sizes, and sources, should be recorded. It is necessary to take photos or video on site and record the other information. Finally, all debris should be collected and removed as far as possible in order to protect the mangrove living environment”.

Then, more attention details are described one by one as follows: “(1)Sampling location, time and area, (2) Materials, size and sources,(3) The density estimation and assessment”.

3) Some details (2) Materials, size and sources have been added in order to indicate clear those difference from other methods. Some sentences or words that duplicate the literature have been deleted to make it more concise. Moreover, a small paragraph is added as follows:

Table 1 provides marine debris classification based on the size, materials and sources. Here, one difference from other methods is that the disposable paper cups (beverage cartons) coated with single-sided PE plastic film or double-sided PE or plastic film are regarded as plastic debris. Because the remaining plastic film after paper fiber being damaged easily will form lots of serious secondary microplastics into the environment. Moreover, wood products refer to mainly those old furniture and ships that are large, hinder the landscape, and may endanger the mangrove forest growth. Here, the wood sticks and furniture fragments that are small and do not endanger the mangroves growth are not considered. 

4) Another small paragraph is modified to indicate clear those importance and advantage of our method, it is as follows:

“Our approach is suitable for all visual debris items encountered in mangrove ecosystem, deposited on the seabed, and/or associated with encrusted/entangled on the branchs/roots. This approach can also provide effective assurance for harmonization of sampling/monitoring methods, identification of sources, attention to natural environmental variability, collation and comparison of data. It can not only be used alone for the special marine debris survey, but also be used in conjunction with the comprehensive mangrove ecosystem investigation. However, there are still some disadvantages, eg., large personnel workload and high manpower consumption (compared with remote sensing method).”

5)2.2. Study area and sampling has changed into 2.2. Application and Practice in Beilun estuary mangrove region .  The established method is applied to the investigation of mangrove marine debris in Beilun estuary:“2.2 Application and Practice in Beilun estuary mangrove region”.

  1. The conclusions also talk about the development of an appropriate method- but it is unclear what the development was as opposed to just doing a method.Additional focus in the introduction is needed with more appropriate conclusions.If the paper was changed to an evaluation of debris in mangrove as opposed to development of a sampling method for debris in mangrove, I think that makes more sense. 

R:

The whole paper should include two part of the development of a sampling method for debris in mangrove” and “an evaluation of debris in Beilun estuary mangrove region. In order to make the development of an appropriate method clear, the section of Introduction has been rewritten as mentioned above. Also, the section of Conclusions  has been modified according the comments and suggestions of reviewers.

 In order to highlight the importance of the two parts, we will also change the second Section of 2. Materials and methods, 2.1. Method Establishment, 2.2. Study area and sampling into “2. Methodology, 2.1. Method Establishment, and 2.2. Application and Practice in Beilun estuary mangrove region.  Such modification makes the paper more logical, makes readers believe that it is important to establish an appropriate and special method, and also makes readers understand the current situation of marine debris in the Beilun estuary mangrove area.

We modify other parts as follows

  1. In Line 147 of Page 4/14, “Note”in Table 1 has been added as follows:

Classification

Specific examples

Materials/Composition

(plastics, styrofoam, wood, paper, metal, rubber, fabric/fiber,glass, other material)*

Plastics

Bags, bottles, cigarette filters, lighters, buckets, lid, spoons, knives and forks, straws, hats, diapers, syringes, fishing lines, nets, floats, safety helmets, feeding bottles, ropes, toys, rings, old plastic boat, etc.;

Styrofoam

Buoys, cups, foam boxes, fast food boxes (plates), etc.;

Glasses

Bottles, fluorescent tubes, bulbs, glass fragments, etc.; 

Metals

metal barrels, beverage cans, metal plates, metal fragments,iron wires, etc.;

Rubbers

balloons, rubber gloves, tires, condoms, etc.;

Fabrics

Clothes, rags, and textile materials, etc.;

Wood products#

Furniture, crates,chopsticks,wooden cases, old wooden boat, , etc.;

Papers$

Paper bags, cardboards, cups, newspapers, etc.;

Others

Other artificial articles and unrecognizable materials

Size

Small-size (<2.5 cm), Medium-size (≥2.5 cm and ≤10 cm), Large-size (>10 cm and ≤1 m) or Oversize (>1 m) debris

Sources

Marine sources

coastal/recreational activity, navigation/fishing activity, medical or sanitary activity, and other disposal source (e.g. wood, ceramics, plastic pieces, glass slices)

Land sources

Note:*The relative persist-ency: food waste < paper < wood < iron < plastic. # Wood products are mainly those furniture and ships that are large, hinder the landscape, and may endanger the mangrove forest growth. $Papers refer to paper products without plastic coating. The disposable paper cups (or beverage cartons) are considered as plastics, because the paper surface is coated with single- or double-sided PE or plastic film.

  1. SI Table 1  has been changed into “Table 3.

1 ) “SI Table 1 ” has been changed into “Table 3. Mass weights and quantities of debris in Shijiao, Jiaodong, Zhushan and Rongshutou transects in Beilun estuary mangrove forest”.

2) The content pattern of the table has been reproduced and adjusted.

3) Some Errors in the table have been modified, e.g., marine debris item, quantity density (item/m2) and Percentage(%) in Zhushan transect (Station ZS-1, ZS-2, and ZS-3 ), the mumblers of “0.200” for Fabrics, “0.612” “3.92”, “61.44”, “32.68”, “1.96” have been changed into “0.008” , “0.420” “5.71”, “89.52”, “1.90”, “2.86” , the number of “50” has been deleted (should be “2”).

 Some Errors in Total-Four transects (Shijiao , Jiaodong , Zhushan, and Rongshutou) have been also modified:  the mumblers of “0.047”  for Fabrics , “0.203” , “11.52”, “60.91”, “2.88”, “1.23” , “23.05” , “0.41”  have been changed into “0.007 ” , “0.163”,  “14.36”, “75.90”, “3.59”, “1.54” , “4.10” , “0.51”  , the number of “56” has been deleted (should be “8”).

In “Figure 3. The mass and quantity densities and the percentages of the debris”, “Abstract”, and  “Results and Discussion”,  the corresponding errors have been modified.

  1. Table 4. Source of marine debris in Beilun estuary mangrove forest region has been added.  In Results and Discussion,  the corresponding errors have been added as follows:

Table 4 provides source of marine debris in Beilun estuary mangrove forest region. For marine debris, the mass percentage of 9.19% and the quantity percentage of 5.64% came from marine activities,viz., navigation/fishing activity. In the land-based debris, the mass percentage of 0.12% and the quantity percentage of 1.54% came from the medical or sanitary activity. More than 90% (both mass and quantity percentages) came from the coastal/recreational activity. More than 50% of the land-based debris were plastics, followed by styrofoam (more than 10%).

  1. SI Table 2  has been changed into “Table 6.

  1. Fix the typos and polish the language as well.
  • In Line 133 OFPage 3/14, Line 260 of Page 8/13, Line 267, 285 and 294 of Page 9/13, Line 345 of Page 10/13, the word “Addationally” has been changed into “Additionally”.
  • In Line 69 of Page 1/14, Line 344 of Page 10/14, the word of “harmonisation”has been changed into “harmonization” .
  • In Line 81 of Page 2/14, In Table 1, Line 143 of Page 4/14, and In Line 259 of Page 8/14, the word of “etc”was changed into “”
  • In Line 127 of Page 3/14, Line 359 of Page 11/14, the word of “enviroment”has been changed into “environment” .
  • In Line 140 of Page 3/14, Line 359 of Page 11/14, the word of “investigatinon”has been changed into “investigation” .
  • In Line 192 of Page 5/14, Line 359 of Page 11/14, the word of “mairne”has been changed into “marine” .
  • In Line 214-215 of Page 6/14, in Figure 2 the words of “(d) Plastic bag and box at Rongshutou”  has been changed into “(d) Beverage boxes and used clothes at Rongshutou” .
  • In Line 216-217 of Page 6-7/14, in Table 2,
  • the words of “small/Big plastic bag”,“clothes”, “ cloths” has been changed into “small/big plastic bag”, “used clothes” .
  • In Line 358-359 of Page 11/14, the word of “recommendation”has been changed into “recommendations” .
  • In Line 362.367 of Page 11/14, the word of “Fangcheng”has been changed into “Fangchenggang city” .

Reviewer 3 Report

The manuscript “Marine debris in the Beilun estuary mangrove forest: Monitoring, Assessment and Implications” seeks to investigate the anthropogenic debris found in  Beilun estuary mangrove  forest region in Fangchenggang,   using the visual and weight approaches.

The manuscript is at an initial stage and needs to be rewritten (improvement in English language and in the sequence in which the information is given). There are many flaws in the methodology presented as well in the interpretation of results and the conclusions.

The methodology must be clearly explained and the new implementations mentioned must be clearly described.

By no means can this manuscript be published as it is.

Abstract

The abstract is too short and without a clear objective.

 “The study tries ….to  understand  the  impact  of  human  activities  and  their  waste  on mangroves…” ???

Please explain in more detail what is the object and what is the methodology used, as no information on the applied methodology is given in the abstract

With respect to the results:

 “All  debris  with  plastics  (>60%),  styrofoam  (>10%),  and  fabrics (>15%) came from human activities.”

The finding that it “came from human activities” is trivial.

Introduction

Please indicate clear at the end of the introduction what is the objective of the study?

Methodology

The first and second paragraph

The discussion of the available methods used in other studies should not be discussed here (this could be in the introduction or the discussion section

2.1. Method Establishment

No systematic method is presented but

“Based  on  Guideline  for  marine  debris  monitoring  and  assessment  (SOA   of  China, 2011),  a  method  using  the  visual  and  weight  method  for  marine  debris  in  mangrove forest  is  used  to  assess  the  debris  coverage  and  their  impacts  on  mangroves.” 

(How does it function and how and why did you modify it?)

“One approach to make  best use  of limited resources is to take advantage of other studies and programmes  (should be programs) where  litter  monitoring  can  be  integrated  (GESAMP,  2019).” 

(How does it function and how and why did you modify it?)

1) Sampling location, time and area

“Sampling  location  (transect  and  station)  and  time  depends  on  the  surrounding enviroment  (should be environment)  conditions  and  the  flood  and  ebb  tides.  Sampling  should  not  always  take place  at  a  constant  location/transect.  Each  transect  include  (should be includes) three  stations,  viz.,  the high-tide  station,  the  medium-tide  station,  and  the  low-tide  station.”

Please write clearly what criteria did you use in order to decide which sampling location, time, and area were best. Please give reasons why did you make these variations.

Equations (1) please use an equation editor  

Results

The table attached as SUPPLEMENTARY MATERIAL  should be integrated into the text (as results), in order to better discuss the findings.

  1. Conclusions

This section has been written without care and needs complete rewriting

YOU WRITE:

“Our stduy is useful for providing recommendation and practical guidance for establishing programmes to monitor and assess the distribution and abundance of mairne plastic debris.”

SHOULD BE

Our study is useful for providing recommendations and practical guidance for establishing programs to monitor and assess the distribution and abundance of marine plastic debris.

YOU WRITE:

In order to protect mangrove forest and reduce the debris, more efforts has been made in Fangcheng city, China. The production, sales and use of ultra-thin bags (<0.025mm) was banned in China on June 1, 2008.

SHOULD BE:

In order to protect mangrove forests and reduce the debris, more efforts have been made in Fangcheng city, China. The production, sales, and use of ultra-thin bags (<0.025mm) was banned in China on June 1, 2008.

The text should respond if the objectives were completed and in which way the methodology may be generalized for other regions in the world and what should be the next step following this study.

The last section on the conclusion should be part d the discussion.

Author Response

Reviewer #3:

Author's Reply to the Review Report (Reviewer 3)

Open Review

English language and style

( ) Extensive editing of English language and style required
(x) Moderate English changes required
( ) English language and style are fine/minor spell check required
( ) I don't feel qualified to judge about the English language and style

Yes

Can be improved

Must be improved

Not applicable

Does the introduction provide sufficient background and include all relevant references?

( )

( )

(x)

( )

Is the research design appropriate?

( )

( )

(x)

( )

Are the methods adequately described?

( )

( )

(x)

( )

Are the results clearly presented?

( )

( )

(x)

( )

Are the conclusions supported by the results?

( )

( )

(x)

( )

Comments and Suggestions for Authors

  1. The manuscript “Marine debris in the Beilun estuary mangrove forest: Monitoring, Assessment and Implications” seeks to investigate the anthropogenic debris found inBeilun estuary mangrove forest region in Fangchenggang, using the visual and weight approaches. 

The manuscript is at an initial stage and needs to be rewritten (improvement in English language and in the sequence in which the information is given). There are many flaws in the methodology presented as well in the interpretation of results and the conclusions.

The methodology must be clearly explained and the new implementations mentioned must be clearly described.

By no means can this manuscript be published as it is.

R:

OK.

The manuscript has be rewritten in the sequence in which the information is given. The whole paper should include two part of the development of a sampling method for debris in mangrove” and “an evaluation of debris in Beilun estuary mangrove region. In order to make the development of an appropriate method clear, the section of Introduction has been rewritten as mentioned above. Also, the section of Conclusions  has been modified according the comments and suggestions of reviewers. 

In order to highlight the importance of the two parts, we will also change the second Section of 2. Materials and methods, 2.1. Method Establishment, 2.2. Study area and sampling into “2. Methodology, 2.1. Method Establishment, and 2.2. Application and Practice in Beilun estuary mangrove region.  Such modification makes the paper more logical, makes readers believe that it is important to establish an appropriate and special method, and also makes readers understand the current situation of marine debris in the Beilun estuary mangrove area.

English language also has been improved.

  1. 2.Abstract

The abstract is too short and without a clear objective. “The study tries  …. to understand the impact of human activities and their waste on mangroves…” ??? Please explain in more detail what is the object and what is the methodology used, as no information on the applied methodology is given in the abstract. With respect to the results:  “All debris with plastics (>60%), styrofoam (>10%), and fabrics (>15%) came from human activities.” The finding that it “came from human activities” is trivial.

R: 

  1. Abstracthas been rewritten.

The Section of Abstract The debris items were investigated in Beilun estuary mangrove forest region in Fangchenggang in 2019. To understand the impact of human activities and their waste on mangroves, some practical programmes, viz, sampling location, time, area, materials, size and sources, data processing, are provided for marine debris using the visual and weight approachs. The mass and quantity densities are 21.123 (2.355~51.760) g/m2 and 0.203 (0.013~0.612) item/m2, respectively. All debris with plastics (>60%), styrofoam (>10%), and fabrics (>15%) came from human activities. The big-size debris is more than 70%. The density and type at Zhushan and Rongshutou near the China-Vietnam border are more than those at Shijiao and Jiaodong. Results suggest that mangrove forest are barriers for the medium-/big land/ocean-based debris, acting as traps for marine debris. Our study is useful for providing recommendation and practical guidance for establishing programmes to monitor and assess the distribution and abundance of marine debris. Also, it can provide important information and support for the mangrove ecosystem protection and restoration near the China-Vietnam border. has been modified as follows:

Here an appropriate approach for marine debris investigation in mangrove forest is developed including some practical programmes, viz., sampling location, time, area, materials, size and sources, data processing. Marine debris method was practiced in Beilun estuary mangrove forest region in Fangchenggang in 2019, viz., the debris items were classified, counted, weighted, recorded, and assessed marine debris pollution to understand the impact of human activities. Results show that the mass density is 21.123 (2.355~51.760) g/m2, and more than 90% came from the land-based human activities. More than 60% of total debris weights is plastics, followed by fabrics (17.91%) and styrofoam (10.07%); the big-size and oversize debris are 76.41% and 13.33%. The quantity density is 0.163 (0.013~0.420) item/m2, and ~95% came from the land-based human activities. More than 75% of total debris items was plastics, followed by styrofoam (14.36%), fabrics (4.10%) and Glasses (3.59%); the big-size, medium-size and oversizedebris are 76.41%, 13.33%, and 10.26%, respectively. Results suggest that mangrove forest are barriers for the medium-/big land/ocean-based debris, acting as traps for marine debris. Our study is useful for providing recommendations and practical guidance for establishing programmes to monitor and assess the distribution and abundance of marine debris. Results show that mangrove areas in Beilun estuary are filled with some plastic debris ( plastics plus styrofoam), and that the density and type at Zhushan and Rongshutou near the China-Vietnam border are more than those at Shijiao and Jiaodong. This study is also expected to not only provide baseline data for the future assessment of Beilun estuary mangroves, but also help China and Vietnam strengthening marine land-based pollution control, and promote coastal wetland and mangrove conservation, marine species conservation and sustainable use.

  1. 3.Introduction,

Please indicate clear at the end of the introduction what is the objective of the study?

R:

OK.

The whole Section of Introduction has been rewritten. Moreover, in order to indicate clear the study objective, the end of the introduction has been rewritten as follows:

Beilun estuary mangrove forest reserve region is located at the southern offshore areas of Gangkou district and Dongxing city of Fangchenggang city, China, bordered Beibu gulf in the southeast and Vietnam in the southwest, and spanned Pearl bay, Jiangping estuary, and Beilun estuary, with a coastline of 105 km and a tidal flat area of 53 km2 [29, 30-33]. Here we provide an approach for marine debris investigation in mangrove forest including some practical programmes, viz., sampling location, time, area, materials, size and sources, data processing. Marine debris method was applied and practiced in Beilun estuary mangrove forest region in 2019, viz., the debris items were classified, counted, weighted, recorded, and assessed the distribution and abundance of marine debris pollution. It is hoped that our study is useful for providing recommendations and practical guidance for monitoring and assessing marine debris in mangrove forest, and for understanding the basic information of mangrove ecological threat factors and to assess the influence of human activities. This study is also expected to not only provide baseline data for the future assessment of marine debris pollution in Beilun estuary mangroves, but also help China and Vietnam strengthening marine land-based pollution control, and promote coastal wetland and mangrove conservation, marine species conservation and sustainable use.

4.Methodology 

The first and second paragraph:The discussion of the available methods used in other studies should not be discussed here (this could be in the introduction or the discussion section.

R:

   OK.

Line 85-line 109, Page 2-3/14, the first and second paragraphs of Methodology “2. Materials and methods” have been transferred to the second and third paragraphs of 1. Introduction .

Our aim is to make it clear that “No appropriate and special method is used for monitoring and assessing marine debris in mangroves forest. ” it is essential to develop an appropriate method of marine debris in mangroves monitoring, to provide effective recommendations and practical guidances”. Also, this paves the way for the second Section of "2. Methodology.  The section modified is as follows:

For marine debris survey, some methods have been successfully established for marine debris monitoring and assessment in marine environment (surface seawater, seabed, and beaches)[8, 13-17]. The approaches/methods comprise the visual method, net method (trawl), diving method [12, 15]. Also, some innovative approaches here include satellite remote sensing [18-20], aerial photography method [20-22] using coupled balloon-assisted photos with in situ mass measurements [23], ortho-photograph from planes[19], and several initiatives using drones [24]. These innovative approaches are particularly useful for detecting larger litter items in dense vegetation (e.g. reed beds), for non- destructive observations in sensitive habitats (e.g. salt marshes) and for remote or inaccessible coastlines [8]. Ongoing monitoring activities can be used to assess the effectiveness of management strategies, and provide an insight when strategies need to be modified for changing conditions. There was a lack of harmonization of sampling methods and attention to natural environmental variability, so that this made the collation and comparison of data problematic [8,13]. The implementation of effective control measures is currently hampered by a lack of consistent monitoring and identification of sources of debris [12].

No appropriate and special method is used for monitoring and assessing marine debris in mangroves forest. Mangroves are usually located in wetland and marsh areas of estuaries and are subjected to tidal influences, so that the debris investigation in mangrove forests (especially in different inter-tidal zones) is much more difficult than those in seawater, seabed, and beaches. The satellite and photography method is not applicable because of the canopy of the mangrove branches, so that the visual and weight methods are still simple and practical. Although marine debris survey in mangrove forest is similar to that in sea water, beach and seabed [8,12-15], there is some differences. Up to now, there are a few studies on marine debris monitoring and assessing in mangroves [9,25], but there is no appropriate and special method. Marine debris pollution (including plastics and microplastics) in mangrove forest regions are actually affected by both natural factors (i.e., hydrodynamics, mangrove height and density, etc.) and human activities (mariculture, tourism and coastal dumping)[26-29]. Thus, it is essential to develop an appropriate method of marine debris in mangroves monitoring, to provide effective recommendations and practical guidances, and to conduct surveys to assess the impact of human activities on coastal zones.

  1. “2.1. Method Establishment”

No systematic method is presented but “Based on Guideline for marine debris monitoring and assessment (SOA of China, 2011), a method using the visual and weight method for marine debris in mangrove forest is used to assess the debris coverage and their impacts on mangroves.”  (How does it function and how and why did you modify it?)

R:

OK.

The guideline for marine debris monitoring and assessment (SOA of China, 2011) is used for monitoring and assessment of marine debris in sea water, seabed, and breach. It is unsuited for monitoring marine debris in mangrove forest. The reasons have been mentioned in the previous “Introduction”.

The sentence of “Based on Guideline for marine debris monitoring and assessment (SOA of China, 2011), a method using the visual and weight method for marine debris in mangrove forest is used to assess the debris coverage and their impacts on mangroves” has been changed into “By referring to the current methods or guidelines for marine debris (in sea water, seabed, and breach) monitoring and assessment[8,13-15],an special approach for marine debris in mangrove forest is developed using the visual and weight methods. ”

6.“One approach to make best use of limited resources is to take advantage of other studies and programmes (should be programs) where litter monitoring can be integrated (GESAMP, 2019).”

 (How does it function and how and why did you modify it?)

R:

     OK.

The sentence of “One approach to make best use of limited resources is to take advantage of other studies and programmes where litter monitoring can be integrated (GESAMP, 2019).” has been deleted.

The word of “programmes” (should be programs) has not been modified. Because that there is  a marked difference in usage and meaning between the two words of “programme” and “program”.  Here programme is a good word to use.

  1. “1) Sampling location, time and area”

“Sampling location (transect and station) and time depends on the surrounding enviroment (should be environment) conditions and the flood and ebb tides. Sampling should not always take place at a constant location/transect. Each transect include (should be includes) three stations, viz., the high-tide station, the medium-tide station, and the low-tide station.”

Please write clearly what criteria did you use in order to decide which sampling location, time, and area were best. Please give reasons why did you make these variations.

R:

OK.

The word of “enviroment (should be environment)  has been modified.

The criteria used in order to decide which sampling location, time, and area has been re-described. In the section of “2.1. Method Establishment”, the important criteria used in order to decide which sampling location, time, and area, is the survey purpose——“Firstly, sampling location (transect and station) and time are drawn up according to the survey purpose”. The section of “1) Sampling location, time and area” has been modified as follows:

Sampling location (transect and station) and time setting can refer to the general specifications or guidelines of mangrove survey or marine inter-tidal zone. Also, sampling location and time setting must consider the surrounding environment conditions and the flood and ebb tides. Sampling should not always take place at a constant location/transect.  Each transect include three stations, viz., the high-tide station, the medium-tide station, and the low-tide station. Random sampling location can also be used. Sampling time must comply with the safety regulations and the tide time because of the complex environment (mud everywhere, twining branches). If the tide rises fast, sampling time duration must be controlled to ensure the personnel security before the rising tide. Additionally, personal protection (to prevent mosquitoes and snake bites) must be prepared; monitoring work must stop when it rains and lightning strikes.

The selection of sampling area (m2) in each station generally follows the following principles: The larger sampling area is, the more representative and reliable the results are. Also, the densities of the debris and mangrove must been considered, as well as  the surrounding environment conditions and Duration between the flood and ebb tides. Debris coverage was estimated in 10 m×10 m block which is divided into 1 m2 quadrats and 20 quadrats from the 100 m2 block were sampled in each site [34]. For ease of operation, if a comprehensive mangrove ecological investigation are carried out, the location, time and area are same as or close to those of debris in the mangrove forsts. Here, the area in each station are often 100 m2. If the density is too small, the area can be appropriately increased. If the density is very high or the environment is unfavorable for the investigation, the area can be reduced but should not be less than 5 m×5 m.

  1. Equations (1) please use an equation editor  

R:

  1.  Equations (1) has been modifiedusingan equation editor.   

9.ResultsThe table attached as “SUPPLEMENTARY MATERIAL”  should be integrated into the text (as results), in order to better discuss the findings.

R: 

OK.

     In order to better discuss the findings, SI Table 1 and Table 2 as results have been integrated into the text.  SI Table 1 was changed into Table 3, and SI Table 2 was changed into Table 5. Table 3 was changed into Table 4.

(1)The size in the Table 2 is reduced.

(2) In Line 375-377 of Page 11/14, Supplementary material of Data has been deleted in text.

10.Conclusions,  This section has been written without care and needs complete rewriting.  

R:  

  1.  

The section of “Conclusions” has been rewritten completely. 

11.YOU WRITE: “Our stduy is useful for providing recommendation and practical guidance for establishing programmes to monitor and assess the distribution and abundance of mairne plastic debris.”  SHOULD BE “Our study is useful for providing recommendations and practical guidance for establishing programmes to monitor and assess the distribution and abundance of marine plastic debris.”

R:  

OK.

In line 357-359 of Page 11/14, the sentence of “Our stduy is useful for providing recommendation and practical guidance for establishing programmes to monitor and assess the distribution and abundance of mairne plastic debris.” has been changed into “Our study is useful for providing recommendations and practical guidance for establishing programmes to monitor and assess the distribution and abundance of marine plastic debris.”  

In the sentence, the words of “recommendation” and “mairne” have been changed into “recommendations” and “marine” . However, the word of “programmes” has not been changed into “programs”, because that there is  a marked difference in usage and meaning between the two words of “programme” and “program”. Here programme is a good word to use.

12.YOU WRITE:  “In order to protect mangrove forest and reduce the debris, more efforts has been made in Fangcheng city, China. The production, sales and use of ultra-thin bags (<0.025mm) was banned in China on June 1, 2008.”  SHOULD BE: “In order to protect mangrove forests and reduce the debris, more efforts have been made in Fangcheng city, China. The production, sales, and use of ultra-thin bags (<0.025mm) was banned in China on June 1, 2008.”

R:

OK.

The sentence of  “In order to protect mangrove forest and reduce the debris, more efforts has been made in Fangcheng city, China. The production, sales and use of ultra-thin bags (<0.025mm) was banned in China on June 1, 2008.”  has been changed into:

 “In order to protect mangrove forest and reduce the debris, more efforts have been made in Fangchenggang city, China. The production, sales and use of ultra-thin bags (<0.025mm) was banned in China on June 1, 2008.”

13.The text should respond if the objectives were completed and in which way the methodology may be generalized for other regions in the world and what should be the next step following this study.

R:

if the objectives were completed and in which way the methodology may be generalized for other regions in the world and what should be the next step following this study.

The Section of Conclusions” has been modified as follows:

“The magnitude and the fate of this pollution are still open questions. A appropriate approach of marine debris in mangrove was established and practiced in the Beilun estuary mangrove region. Our results show that mangroves act mostly as a barrier of the big-size debris, because it can prevent both the land-based debris from dispersing into the sea and the ocean-based debris from invading the terrestrial environment. It is hoped that our methodology may be generalized for other regions in the world, and that it is helpful to promote the standardized monitoring of marine debris in mangrove ecosystem to assess the impact of human activities on mangroves. This study is also expected to help biodiversity friendly practices and approaches promoted for conservation and sustainable use of threatened ecosystems and species in mangrove ecosystem. 

The vast mangrove ecosystem in the Beilun estuary provides food and habitat for a variety of benthos and birds. We must clearly recognize that mangrove areas in Beilun estuary are filled with some plastic debris ( plastics plus styrofoam: more than 70% Ms- p.c and 90% Qt-p.c.) due to human activities. In Beilun estuary mangrove region, marine debris pollution is actually an international problem. Because the Beilun estuary region is a land connection for economic cooperation and logistics between China and ASEAN. Dongxi in Fangchenggang (China) is only separated from Mong Cai (Vietnam) by the Beilun River. With the acceleration of industrialization and urbanization, and the rapid development of economy and society, the mangroves are facing serious threats due to urban expansion, coastal development, high-intensity human interference, and other factors. Our results is helpful to promote the supervision and control of marine debris, especially plastics, in the mangrove region of the Beilun estuary of China and Vietnam. The findings of this study provide baseline data for marine debris including plastic pollution, and assist in prioritizing future plastic debris monitoring and mitigation strategies.”

14.The last section on the conclusion should be part of the discussion.

R:

  1.  The last section on the conclusion has been transferred to be part of the discussion.“ 3.1. Space distribution of marine debris” in 3. Results and Discussion

We modify other parts as follows

  1. In Line 147 of Page 4/14, “Note”in Table 1 has been added as follows:

Classification

Specific examples

Materials/Composition

(plastics, styrofoam, wood, paper, metal, rubber, fabric/fiber,glass, other material)*

Plastics

Bags, bottles, cigarette filters, lighters, buckets, lid, spoons, knives and forks, straws, hats, diapers, syringes, fishing lines, nets, floats, safety helmets, feeding bottles, ropes, toys, rings, old plastic boat, etc.;

Styrofoam

Buoys, cups, foam boxes, fast food boxes (plates), etc.;

Glasses

Bottles, fluorescent tubes, bulbs, glass fragments, etc.; 

Metals

metal barrels, beverage cans, metal plates, metal fragments,iron wires, etc.;

Rubbers

balloons, rubber gloves, tires, condoms, etc.;

Fabrics

Clothes, rags, and textile materials, etc.;

Wood products#

Furniture, crates,chopsticks,wooden cases, old wooden boat, , etc.;

Papers$

Paper bags, cardboards, cups, newspapers, etc.;

Others

Other artificial articles and unrecognizable materials

Size

Small-size (<2.5 cm), Medium-size (≥2.5 cm and ≤10 cm), Large-size (>10 cm and ≤1 m) or Oversize (>1 m) debris

Sources

Marine sources

coastal/recreational activity, navigation/fishing activity, medical or sanitary activity, and other disposal source (e.g. wood, ceramics, plastic pieces, glass slices)

Land sources

Note:*The relative persist-ency: food waste < paper < wood < iron < plastic. # Wood products are mainly those furniture and ships that are large, hinder the landscape, and may endanger the mangrove forest growth. $Papers refer to paper products without plastic coating. The disposable paper cups (or beverage cartons) are considered as plastics, because the paper surface is coated with single- or double-sided PE or plastic film.

  1. SI Table 1  has been changed into “Table 3.

1 ) “SI Table 1 ” has been changed into “Table 3. Mass weights and quantities of debris in Shijiao, Jiaodong, Zhushan and Rongshutou transects in Beilun estuary mangrove forest”.

2) The content pattern of the table has been reproduced and adjusted.

3) Some Errors in the table have been modified, e.g., marine debris item, quantity density (item/m2) and Percentage(%) in Zhushan transect (Station ZS-1, ZS-2, and ZS-3 ), the mumblers of “0.200” for Fabrics, “0.612” “3.92”, “61.44”, “32.68”, “1.96” have been changed into “0.008” , “0.420” “5.71”, “89.52”, “1.90”, “2.86” , the number of “50” has been deleted (should be “2”).

 Some Errors in Total-Four transects (Shijiao , Jiaodong , Zhushan, and Rongshutou) have been also modified:  the mumblers of “0.047”  for Fabrics , “0.203” , “11.52”, “60.91”, “2.88”, “1.23” , “23.05” , “0.41”  have been changed into “0.007 ” , “0.163”,  “14.36”, “75.90”, “3.59”, “1.54” , “4.10” , “0.51”  , the number of “56” has been deleted (should be “8”).

In “Figure 3. The mass and quantity densities and the percentages of the debris”, “Abstract”, and  “Results and Discussion”,  the corresponding errors have been modified.

  1. Table 4. Source of marine debris in Beilun estuary mangrove forest region has been added.  In Results and Discussion,  the corresponding errors have been added as follows:

Table 4 provides source of marine debris in Beilun estuary mangrove forest region. For marine debris, the mass percentage of 9.19% and the quantity percentage of 5.64% came from marine activities,viz., navigation/fishing activity. In the land-based debris, the mass percentage of 0.12% and the quantity percentage of 1.54% came from the medical or sanitary activity. More than 90% (both mass and quantity percentages) came from the coastal/recreational activity. More than 50% of the land-based debris were plastics, followed by styrofoam (more than 10%).

  1. SI Table 2  has been changed into “Table 6.

  1. Fix the typos and polish the language as well.
  • In Line 133 OFPage 3/14, Line 260 of Page 8/13, Line 267, 285 and 294 of Page 9/13, Line 345 of Page 10/13, the word “Addationally” has been changed into “Additionally”.
  • In Line 69 of Page 1/14, Line 344 of Page 10/14, the word of “harmonisation”has been changed into “harmonization” .
  • In Line 81 of Page 2/14, In Table 1, Line 143 of Page 4/14, and In Line 259 of Page 8/14, the word of “etc”was changed into “”
  • In Line 127 of Page 3/14, Line 359 of Page 11/14, the word of “enviroment”has been changed into “environment” .
  • In Line 140 of Page 3/14, Line 359 of Page 11/14, the word of “investigatinon”has been changed into “investigation” .
  • In Line 192 of Page 5/14, Line 359 of Page 11/14, the word of “mairne”has been changed into “marine” .
  • In Line 214-215 of Page 6/14, in Figure 2 the words of “(d) Plastic bag and box at Rongshutou”  has been changed into “(d) Beverage boxes and used clothes at Rongshutou” .
  • In Line 216-217 of Page 6-7/14, in Table 2,
  • the words of “small/Big plastic bag”,“clothes”, “ cloths” has been changed into “small/big plastic bag”, “used clothes” .
  • In Line 358-359 of Page 11/14, the word of “recommendation”has been changed into “recommendations” .
  • In Line 362.367 of Page 11/14, the word of “Fangcheng”has been changed into “Fangchenggang city” .

Round 2

Reviewer 2 Report

No additional Comments

Author Response

Open Review 2#

English language and style

( ) Extensive editing of English language and style required
(x) Moderate English changes required
( ) English language and style are fine/minor spell check required
( ) I don't feel qualified to judge about the English language and style

Yes

Can be improved

Must be improved

Not applicable

Does the introduction provide sufficient background and include all relevant references?

(x)

( )

( )

( )

Is the research design appropriate?

(x)

( )

( )

( )

Are the methods adequately described?

(x)

( )

( )

( )

Are the results clearly presented?

(x)

( )

( )

( )

Are the conclusions supported by the results?

(x)

( )

( )

( )

(x) Moderate English changes required

Comments and Suggestions for Authors

No additional Comments

  1. Moderate English changes required

R:

   OK.

Other parts has been modified as follows:

  1. The abstracthas been modified as follows:

A modified approach for marine debris investigation in mangrove forests is developed including some practical programs, viz., sampling location, time, area, materials, size and sources, data processing. The marine debris method was practiced in Beilun estuary mangrove forest region in Fangchenggang in 2019, viz., the debris items were classified, counted, weighed, recorded, and assessed marine debris pollution to understand the impact of human activities. Results show that the mass density is 21.123 (2.355~51.760) g/m2, and more than 90% came from the land[1]based human activities. More than 60% of total debris weights is (are) plastics, followed by fabrics (17.91%) and styrofoam (10.07%); the big-size and oversize debris are 76.41% and 13.33%. The quantity density is 0.163 (0.013~0.420) item/m2, and ~95% came from land-based human activities. More than 75% of total debris items were plastics, followed by styrofoam (14.36%), fabrics (4.10%) and, glass(3.59%); the big-size, medium-size and oversize debris are 76.41%, 13.33%, and 10.26%, respectively. Results suggest that mangrove forests are barriers for the medium-/big land/ocean[1]based debris, acting as traps for marine debris. Our study provide recommendations and practical guidance for establishing programs to monitor and assess the distribution and abundance of marine debris. Results show that mangrove areas in Beilun estuary are filled with some plastic debris (plastics plus styrofoam), and that the density and type at Zhushan and Rongshutou near the China-Vietnam border are more than those at Shijiao and Jiaodong. The results of this study are also expected to not only provide baseline data for the future assessment of Beilun estuary mangroves, but also to help China and Vietnam strengthening marine land-based pollution control, and promote coastal wetland and mangrove conservation, marine species conservation, and sustainable use.

  1. In the manuscript, the word of programmeshas been changed into “programs”.

  1. Introduction, Paragraph 1, Line 7-8 of Page 2, the sentence“Martin et al.(2019)[9] demonstrated that mangrove forest act as sinks for anthropogenic debris...”has been Changed to“Martin et al. (2019)[9] demonstrated that mangrove forests act as sinks for anthropogenic debris...

  1. Introduction, Paragraph 3, Line 34-35 of Page 2, the sentence“No appropriate and special approachis used for monitoring and assessing marine debris in mangroves forest.”has been Changed to“No appropriate and special approach is used for monitoring and assessing marine debris in mangroves forests.

1. Introduction, Paragraph 3, Line 34-35 of Page 2, the sentence“Although marine debris survey in mangrove forest is similar to that in sea water, beach and seabed [8,12-15]”has been Changed to“Although marine debris survey in mangrove forests is similar to that in sea water, beach and seabed [8,12-15]

1. Introduction, Paragraph 4, Line 1-2 of Page 3, the sentence“Here we provide an approach for marine debris investigation in mangrove forest including some practical programmes,”has been Changed to“Here we provide a modified approach for marine debris investigation in mangrove forests including some practical programs,

1. Introduction, Paragraph 4, Line 8 of Page 3, the sentence“mangrove forest, and for understanding the basic information of mangrove ecological ”has been Changed to“mangrove forests, and for understanding the basic information of mangrove ecological

1. Introduction, Paragraph 4, Line 13 of Page 3, the sentence“...marine species conservation and sustainable use.”has been Changed to“...marine species conservation, and sustainable use. 

  1. Methodology”, Line 19-20 of Page 3, the sentence“an special approach for marine debris in mangrove forest is developed using the visual and weight methods. ”has been Changed to“an special approach for marine debris in mangrove forestsis developed using the visual and weight methods.

2. Methodology”, Line 35 of Page 3, the sentence“constant location/transect.  Each transect include three stations, viz., the high-tide ”has been Changed to“constant location/transect. Each transect include three stations, viz., the high-tide ”. Another Space is removed

  1. In Table 1, The noteNote:*The relative persist-ency: food waste < paper < wood < iron < plastic. # Wood products are mainly those furniture and ships that are large, hinder the landscape, and may endanger the mangrove forest  $Papers refer to paper products without plastic coating. The disposable paper cups (or beverage cartons) are considered as plastics, because the paper surface is coated with single- or double-sided PE or plastic film.  ”has been Changed to“Note:*The relative persist-ency: food waste < paper < wood < iron < plastic. # Wood products are mainly those furniture and ships that are large, hinder the landscape, and may endanger the mangrove forest growth. $Papers refer to paper products without plastic coating, which is big, degradable, and endanger the organisms in the mangrove forests. The disposable paper cups (or beverage cartons) are considered as plastics, because the paper surface is coated with single- or double-sided PE or plastic film. 

  1. Methodology”,Line 7-12 of Page 4, the sentenc“Here, one difference from other methods is that the disposable paper cups (beverage cartons) coated with single-sided PE plastic film or double-sided PE or plastic film are regarded as plastic debris. Because the remaining plastic film after paper fiber being damaged easily will form lots of serious secondary microplastics into the environment.”has been Changed to“Here, one difference from other methods is that the disposable paper cups (beverage cartons) coated with single-sided PE plastic film or double-sided PE or plastic film are regarded as plastic debris. Because the remaining plastic film after paper fiber being damaged easily will form lots of serious secondary microplastics into the environment. Papers refer to paper products without plastic coating, which is big, degradable, and endangerthe organisms in the mangrove forests. So the papers that is very easy to degrade in the slush are not considered.

  1. Methodology”, Line 4-5 of Page 6, the sentenc“Table 3, Figure 3 and4 provide the mass and quantity densities and percentages of the debris in Shijiao, Jiaodong, Zhushan and Rongshutou transects, respectively. ”has been Changed to“Table 3, Figure 3(a), Figure 3(b), and Figure 4 provide the mass and quantity densities and percentages of the debris in Shijiao, Jiaodong, Zhushan and Rongshutou transects, respectively. 

  1. In Table 4, the pattern of the table has been modifiedas follows:

Sources

Types

Ms-p.c. (%)

Qt-p.c.

(%)

Debris items

Land-based

Coastal/Recreational

activity

Styrofoam

10.07%

14.36%

Particles,foam fast food box,etc.

Plastics

53.79%

68.72%

Plastic film, plastic bag (woven bag, plastic reticule, packing bag), plastic bottle/box/cup (mineral water bottle, drink bottle/ food box, disposal cup), fruits packing net, packing rope, pipe, cellphone, etc.

Glasses

5.92%

3.59%

Beer bottle, glass bottle/ cup

Metals

1.03%

1.54%

ring-pull can, metal food bottle, motorbike pedal

Rubbers

17.91%

4.10%

Rubber tyre

Fabrics

1.97%

0.51%

Used clothes, shoes,  pants,cotton gloves

Total

90.69%

92.82%

-

Land-based

Medical or sanitary activity

Plastics

0.12%

1.54%

Plastic drug board

Marine

Navigation/fishing activity

Plastics

9.19%

5.64%

Fishing nets, broken buoy, rope

  1. 1. Space distribution of marine debris, Line 4-7 of Page 9, the sentencn four transects, the order of debrismass/quantity densities was as follows: Zhushan > Rongshutou > Shijiao > Jiaodong. The types of debris in Rongshutou and Zhushan transects were more than those in Shijiao and Jiaodong transects.”has been Changed to“In four transects, the order of debris mass/quantity densities is as follows: Zhushan > Rongshutou > Shijiao > Jiaodong. The types of debris in Rongshutou and Zhushan transects are more than those in Shijiao and Jiaodong transects.

  1. 1. Space distribution of marine debris, Paragraph 1, Line 24-25 of Page 9, the sentence“In order to protect mangrove forest and reduce the debris, more efforts had been made in Fangchenggang city, China.”has been Changed to“In order to protect mangrove forests and reduce the debris, more efforts had been made in Fangchenggang city, China.

  1. In Table 6, the titleTable  Mass weights and quantity of marine debris items by size in Beilun estuary mangrove forest” has been modified “Table 6. Mass weights and quantity of marine debris items by sizes.”

  1. 4. Debris distribution influenced by flood-ebb fluctuations, Line 13-17 of Page 11, the sentence“No small-size debris, but the large-debris (the medium-/big size/oversize) are found in the mangrove forest. These debris are mainly light-weighted floating plastic debris (plastics bags and styrofoam). Our results support that mangrove forests are traps [9] for the medium-/big-size, land-/ocean-based floating debris. Additionally, our results suggest that the pneumatophores of mangrove forest act as a filter,”has been Changed to“No small-size debris, but the large-debris (the medium-/big size/oversize) are found in the mangrove forest region. These debris are mainly light-weighted floating plastic debris (plastics bags and styrofoam). Our results support that mangrove forests are traps [9] for the medium-/big-size, land-/ocean-based floating debris. Additionally, our results suggest that the pneumatophores of mangrove forests act as a filter,

  1. 4. Debris distribution influenced by flood-ebb fluctuations, Line 13-17 of Page 11, the sentence“If styrofoam is also considered as plastic debris in this study, the percentage of total plastic debris was more than 70% (the mass percentage of 73.17% and the quantity percentage of 90.25%).”has been Changed to“If styrofoam is also considered as plastic debris in this study, the percentage of total plastic debris is more than 70% (the mass percentage of 73.17% and the quantity percentage of 90.25%).

  1. Conclusions, Line 1-4 of Page 12, the sentence“The magnitude and the fate of this pollution are still open questions. A appropriate approach of marine debris in mangrove was established and practiced in the Beilun estuary mangrove region. has been Changed to“The magnitude and the fate of this pollution are still open questions. An appropriate modified approach of marine debris in mangrove is established and practiced in the Beilun estuary mangrove region.

Reviewer 3 Report

Authors have modified much of the text which now sounds clearer, and the observations have been attended to and corrected, but the English language needs improvement and should be revised by a native English speaker:

For example the abstract:

Abstract: Here an appropriate approach (A modified approach) for marine debris investigation in mangrove forest (forests) is developed including some practical programmes (programs), viz., sampling location, time, area, materials, size and sources, data processing. (The) Marine debris method was practiced in Beilun estuary mangrove forest region in Fangchenggang in 2019, viz., the debris items were classified, counted, weighted (weighed), recorded, and assessed marine debris pollution to understand the impact of human activities. Results show that the mass density is 21.123 (2.355~51.760) g/m2, and more than 90% came from the land[1]based human activities. More than 60% of total debris weights is (are) plastics, followed by fabrics (17.91%) and styrofoam (10.07%); the big-size and oversize debris are 76.41% and 13.33%. The quantity density is 0.163 (0.013~0.420) item/m2, and ~95% came from the land-based human activities. More than 75% of total debris items was (were) plastics, followed by styrofoam (14.36%), fabrics (4.10%) and, Glasses (glass) (3.59%); the big-size, medium-size and oversizedebris (oversize debris) are 76.41%, 13.33%, and 10.26%, respectively. Results suggest that mangrove forest are (forests are) barriers for the medium-/big land/ocean[1]based debris, acting as traps for marine debris. Our study is useful for providing recommendations and practical guidance for establishing programmes (programs) to monitor and assess the distribution and abundance of marine debris. Results show that mangrove areas in Beilun estuary are filled with some plastic debris ( plastics plus styrofoam), and that the density and type at Zhushan and Rongshutou near the China-Vietnam border are more than those at Shijiao and Jiaodong. (The results of this) This study is (are) also expected to not only provide baseline data for the future assessment of Beilun estuary man[1]groves, but also (to) help China and Vietnam strengthening marine land-based pollution control, and promote coastal wetland and mangrove conservation, marine species conservation(,) and sustainable use.

Figure 1. Transects and stations of the debris monitoring in Beilun estuary mangrove forest region

The quality of the figure is low and no information is given on what the different colors mean.

Figure 2. The in-situ monitoring and collection of the debris in mangrove forest.

The quality of the figure is low.

Figure 3. The mass and quantity densities and the percentages of the debris.

The quality of the figure is low.

Figure 4. The percentages of medium-size, big-size and oversize debris

The quality of the figure is very low.

In its present form, the manuscript should not be published.

Author Response

Open Review 3#

English language and style

( ) Extensive editing of English language and style required
(x) Moderate English changes required
( ) English language and style are fine/minor spell check required
( ) I don't feel qualified to judge about the English language and style

Yes

Can be improved

Must be improved

Not applicable

Does the introduction provide sufficient background and include all relevant references?

( )

(x)

( )

( )

Is the research design appropriate?

( )

(x)

( )

( )

Are the methods adequately described?

( )

(x)

( )

( )

Are the results clearly presented?

( )

(x)

( )

( )

Are the conclusions supported by the results?

( )

(x)

( )

( )

Comments and Suggestions for Authors

  1. Authors have modified much of the text which now sounds clearer, and the observations have been attended to and corrected, but the English language needs improvement and should be revised by a native English speaker:

For example the abstract:

Abstract: Here an appropriate approach (A modified approach) for marine debris investigation in mangrove forest (forests) is developed including some practical programmes (programs), viz., sampling location, time, area, materials, size and sources, data processing. (The) Marine debris method was practiced in Beilun estuary mangrove forest region in Fangchenggang in 2019, viz., the debris items were classified, counted, weighted (weighed), recorded, and assessed marine debris pollution to understand the impact of human activities. Results show that the mass density is 21.123 (2.355~51.760) g/m2, and more than 90% came from the land[1]based human activities. More than 60% of total debris weights is (are) plastics, followed by fabrics (17.91%) and styrofoam (10.07%); the big-size and oversize debris are 76.41% and 13.33%. The quantity density is 0.163 (0.013~0.420) item/m2, and ~95% came from the land-based human activities. More than 75% of total debris items was (were) plastics, followed by styrofoam (14.36%), fabrics (4.10%) and, Glasses (glass) (3.59%); the big-size, medium-size and oversizedebris (oversize debris) are 76.41%, 13.33%, and 10.26%, respectively. Results suggest that mangrove forest are (forests are) barriers for the medium-/big land/ocean[1]based debris, acting as traps for marine debris. Our study is useful for providing recommendations and practical guidance for establishing programmes (programs) to monitor and assess the distribution and abundance of marine debris. Results show that mangrove areas in Beilun estuary are filled with some plastic debris ( plastics plus styrofoam), and that the density and type at Zhushan and Rongshutou near the China-Vietnam border are more than those at Shijiao and Jiaodong. (The results of this) This study is (are) also expected to not only provide baseline data for the future assessment of Beilun estuary man[1]groves, but also (to) help China and Vietnam strengthening marine land-based pollution control, and promote coastal wetland and mangrove conservation, marine species conservation(,) and sustainable use.

R:

   OK.

The abstract has been modified as follows:

1A modified approach for marine debris investigation in mangrove forests is developed including some practical programs, viz., sampling location, time, area, materials, size and sources, data processing. The marine debris method was practiced in Beilun estuary mangrove forest region in Fangchenggang in 2019, viz., the debris items were classified, counted, weighed, recorded, and assessed marine debris pollution to understand the impact of human activities. Results show that the mass density is 21.123 (2.355~51.760) g/m2, and more than 90% came from the land[1]based human activities. More than 60% of total debris weights is (are) plastics, followed by fabrics (17.91%) and styrofoam (10.07%); the big-size and oversize debris are 76.41% and 13.33%. The quantity density is 0.163 (0.013~0.420) item/m2, and ~95% came from land-based human activities. More than 75% of total debris items were plastics, followed by styrofoam (14.36%), fabrics (4.10%) and, glass(3.59%); the big-size, medium-size and oversize debris are 76.41%, 13.33%, and 10.26%, respectively. Results suggest that mangrove forests are barriers for the medium-/big land/ocean[1]based debris, acting as traps for marine debris. Our study provide recommendations and practical guidance for establishing programs to monitor and assess the distribution and abundance of marine debris. Results show that mangrove areas in Beilun estuary are filled with some plastic debris (plastics plus styrofoam), and that the density and type at Zhushan and Rongshutou near the China-Vietnam border are more than those at Shijiao and Jiaodong. The results of this study are also expected to not only provide baseline data for the future assessment of Beilun estuary mangroves, but also to help China and Vietnam strengthening marine land-based pollution control, and promote coastal wetland and mangrove conservation, marine species conservation, and sustainable use.

2The word of “programmes” has been changed into “programs”.

  1. Figure 1. Transects and stations of the debris monitoring in Beilun estuary mangrove forest regionThe quality of the figure is low and no information is given on what the different colors mean.

R:

OK.

The abstract has been modified carefully according to the reviewer’s comment.

  1. Figure 2. The in-situ monitoring and collection of the debris in mangrove forestThe quality of the figure is low.

R:

    OK.

The abstract has been modified carefully according to the reviewer’s comment.

Another photo has been added.

  1. Figure 3. The mass and quantity densities and the percentages of the debrisThe quality of the figure is low.

R:

    OK.

The abstract has been modified carefully according to the reviewer’s comment.

Figure 2 becomes changed to two Figures: Figures 2 (A) and 2 (B). the title has been modified “Figure 3(a). The mass (A) and quantity(B) densities of the debris.” and “Figure 3(b). The mass (A) and quantity(B) percentages of the debris.”

  1. Figure 4. The percentages of medium-size, big-size and oversize debris——The quality of the figure is very low.

R:

    OK.

The abstract has been modified carefully according to the reviewer’s comment.

  1. In its present form, the manuscript should not be published.

R:

OK.

The abstract has been modified carefully according to the reviewer’s comment as described above.

Other parts has been modified as follows

  1. Introduction, Paragraph 1, Line 7-8 of Page 2, the sentence“Martin et al.(2019)[9] demonstrated that mangrove forest act as sinks for anthropogenic debris...”has been Changed to“Martin et al. (2019)[9] demonstrated that mangrove forests act as sinks for anthropogenic debris...

  1. Introduction, Paragraph 3, Line 34-35 of Page 2, the sentence“No appropriate and special approachis used for monitoring and assessing marine debris in mangroves forest.”has been Changed to“No appropriate and special approach is used for monitoring and assessing marine debris in mangroves forests.

1. Introduction, Paragraph 3, Line 34-35 of Page 2, the sentence“Although marine debris survey in mangrove forest is similar to that in sea water, beach and seabed [8,12-15]”has been Changed to“Although marine debris survey in mangrove forests is similar to that in sea water, beach and seabed [8,12-15]

1. Introduction, Paragraph 4, Line 1-2 of Page 3, the sentence“Here we provide an approach for marine debris investigation in mangrove forest including some practical programmes,”has been Changed to“Here we provide a modified approach for marine debris investigation in mangrove forests including some practical programs,

1. Introduction, Paragraph 4, Line 8 of Page 3, the sentence“mangrove forest, and for understanding the basic information of mangrove ecological ”has been Changed to“mangrove forests, and for understanding the basic information of mangrove ecological

1. Introduction, Paragraph 4, Line 13 of Page 3, the sentence“...marine species conservation and sustainable use.”has been Changed to“...marine species conservation, and sustainable use. 

  1. Methodology”, Line 19-20 of Page 3, the sentence“an special approach for marine debris in mangrove forest is developed using the visual and weight methods. ”has been Changed to“an special approach for marine debris in mangrove forestsis developed using the visual and weight methods.

2. Methodology”, Line 35 of Page 3, the sentence“constant location/transect.  Each transect include three stations, viz., the high-tide ”has been Changed to“constant location/transect. Each transect include three stations, viz., the high-tide ”. Another Space is removed

  1. In Table 1, The noteNote:*The relative persist-ency: food waste < paper < wood < iron < plastic. # Wood products are mainly those furniture and ships that are large, hinder the landscape, and may endanger the mangrove forest  $Papers refer to paper products without plastic coating. The disposable paper cups (or beverage cartons) are considered as plastics, because the paper surface is coated with single- or double-sided PE or plastic film.  ”has been Changed to“Note:*The relative persist-ency: food waste < paper < wood < iron < plastic. # Wood products are mainly those furniture and ships that are large, hinder the landscape, and may endanger the mangrove forest growth. $Papers refer to paper products without plastic coating, which is big, degradable, and endanger the organisms in the mangrove forests. The disposable paper cups (or beverage cartons) are considered as plastics, because the paper surface is coated with single- or double-sided PE or plastic film. 

  1. Methodology”,Line 7-12 of Page 4, the sentenc“Here, one difference from other methods is that the disposable paper cups (beverage cartons) coated with single-sided PE plastic film or double-sided PE or plastic film are regarded as plastic debris. Because the remaining plastic film after paper fiber being damaged easily will form lots of serious secondary microplastics into the environment.”has been Changed to“Here, one difference from other methods is that the disposable paper cups (beverage cartons) coated with single-sided PE plastic film or double-sided PE or plastic film are regarded as plastic debris. Because the remaining plastic film after paper fiber being damaged easily will form lots of serious secondary microplastics into the environment. Papers refer to paper products without plastic coating, which is big, degradable, and endangerthe organisms in the mangrove forests. So the papers that is very easy to degrade in the slush are not considered.

  1. Methodology”, Line 4-5 of Page 6, the sentenc“Table 3, Figure 3 and4 provide the mass and quantity densities and percentages of the debris in Shijiao, Jiaodong, Zhushan and Rongshutou transects, respectively. ”has been Changed to“Table 3, Figure 3(a), Figure 3(b), and Figure 4 provide the mass and quantity densities and percentages of the debris in Shijiao, Jiaodong, Zhushan and Rongshutou transects, respectively. 

  1. In Table 4, the pattern of the table has been modifiedas follows:

Sources

Types

Ms-p.c. (%)

Qt-p.c.

(%)

Debris items

Land-based

Coastal/Recreational

activity

Styrofoam

10.07%

14.36%

Particles,foam fast food box,etc.

Plastics

53.79%

68.72%

Plastic film, plastic bag (woven bag, plastic reticule, packing bag), plastic bottle/box/cup (mineral water bottle, drink bottle/ food box, disposal cup), fruits packing net, packing rope, pipe, cellphone, etc.

Glasses

5.92%

3.59%

Beer bottle, glass bottle/ cup

Metals

1.03%

1.54%

ring-pull can, metal food bottle, motorbike pedal

Rubbers

17.91%

4.10%

Rubber tyre

Fabrics

1.97%

0.51%

Used clothes, shoes,  pants,cotton gloves

Total

90.69%

92.82%

-

Land-based

Medical or sanitary activity

Plastics

0.12%

1.54%

Plastic drug board

Marine

Navigation/fishing activity

Plastics

9.19%

5.64%

Fishing nets, broken buoy, rope

  1. 1. Space distribution of marine debris, Line 4-7 of Page 9, the sentencn four transects, the order of debrismass/quantity densities was as follows: Zhushan > Rongshutou > Shijiao > Jiaodong. The types of debris in Rongshutou and Zhushan transects were more than those in Shijiao and Jiaodong transects.”has been Changed to“In four transects, the order of debris mass/quantity densities is as follows: Zhushan > Rongshutou > Shijiao > Jiaodong. The types of debris in Rongshutou and Zhushan transects are more than those in Shijiao and Jiaodong transects.

  1. 1. Space distribution of marine debris, Paragraph 1, Line 24-25 of Page 9, the sentence“In order to protect mangrove forest and reduce the debris, more efforts had been made in Fangchenggang city, China.”has been Changed to“In order to protect mangrove forests and reduce the debris, more efforts had been made in Fangchenggang city, China.

  1. In Table 6, the titleTable  Mass weights and quantity of marine debris items by size in Beilun estuary mangrove forest” has been modified “Table 6. Mass weights and quantity of marine debris items by sizes.”

  1. 4. Debris distribution influenced by flood-ebb fluctuations, Line 13-17 of Page 11, the sentence“No small-size debris, but the large-debris (the medium-/big size/oversize) are found in the mangrove forest. These debris are mainly light-weighted floating plastic debris (plastics bags and styrofoam). Our results support that mangrove forests are traps [9] for the medium-/big-size, land-/ocean-based floating debris. Additionally, our results suggest that the pneumatophores of mangrove forest act as a filter,”has been Changed to“No small-size debris, but the large-debris (the medium-/big size/oversize) are found in the mangrove forest region. These debris are mainly light-weighted floating plastic debris (plastics bags and styrofoam). Our results support that mangrove forests are traps [9] for the medium-/big-size, land-/ocean-based floating debris. Additionally, our results suggest that the pneumatophores of mangrove forests act as a filter,

  1. 4. Debris distribution influenced by flood-ebb fluctuations, Line 13-17 of Page 11, the sentence“If styrofoam is also considered as plastic debris in this study, the percentage of total plastic debris was more than 70% (the mass percentage of 73.17% and the quantity percentage of 90.25%).”has been Changed to“If styrofoam is also considered as plastic debris in this study, the percentage of total plastic debris is more than 70% (the mass percentage of 73.17% and the quantity percentage of 90.25%).

  1. Conclusions, Line 1-4 of Page 12, the sentence“The magnitude and the fate of this pollution are still open questions. A appropriate approach of marine debris in mangrove was established and practiced in the Beilun estuary mangrove region. has been Changed to“The magnitude and the fate of this pollution are still open questions. A appropriate modified approach of marine debris in mangrove is established and practiced in the Beilun estuary mangrove region.
